# Wasserstein distributional robustness of neural networks

**Xingjian Bai**
Department of Computer Science
University of Oxford, UK
xingjian.bai@sjc.ox.ac.uk

**Guangyi He**
Mathematical Institute
University of Oxford, UK
guangyihe2002@outlook.com

**Yifan Jiang**
Mathematical Institute
University of Oxford, UK
yifan.jiang@maths.ox.ac.uk

**Jan Obłój**[*]
Mathematical Institute
University of Oxford, UK
jan.obloj@maths.ox.ac.uk

## Abstract

Deep neural networks are known to be vulnerable to adversarial attacks (AA). For an image recognition task, this means that a small perturbation of the original can result in the image being misclassified. Design of such attacks as well as methods of adversarial training against them are subject of intense research. We re-cast the problem using techniques of Wasserstein distributionally robust optimization (DRO) and obtain novel contributions leveraging recent insights from DRO sensitivity analysis. We consider a set of distributional threat models. Unlike the traditional pointwise attacks, which assume a uniform bound on perturbation of each input data point, distributional threat models allow attackers to perturb inputs in a non-uniform way. We link these more general attacks with questions of out-of-sample performance and Knightian uncertainty. To evaluate the distributional robustness of neural networks, we propose a first-order AA algorithm and its multistep version. Our attack algorithms include Fast Gradient Sign Method (FGSM) and Projected Gradient Descent (PGD) as special cases. Furthermore, we provide a new asymptotic estimate of the adversarial accuracy against distributional threat models. The bound is fast to compute and first-order accurate, offering new insights even for the pointwise AA. It also naturally yields out-of-sample performance guarantees. We conduct numerical experiments on CIFAR-10, CIFAR-100, ImageNet datasets using DNNs on RobustBench to illustrate our theoretical results. Our code is available at https://github.com/JanObloj/W-DRO-Adversarial-Methods.

## 1 Introduction

Model uncertainty is a ubiquitous phenomenon across different fields of science. In decision theory and economics, it is often referred to as the *Knightian uncertainty* (Knight, 1921), or the *unknown unknowns*, to distinguish it from the *risk* which stems from the randomness embedded by design in the scientific process, see Hansen and Marinacci (2016) for an overview. Transcribing to the context of data science, risk refers to the randomness embedded in a training by design, e.g., through random initialization, drop-outs etc., and uncertainty encompasses the extent to which the dataset is an adequate description of reality. *Robustness*, the ability to perform well under uncertainty, thus relates to several themes in ML including adversarial attacks, out-of-sample performance and

---

[*]Corresponding author. www.maths.ox.ac.uk/people/jan.obloj

37th Conference on Neural Information Processing Systems (NeurIPS 2023).

out-of-distribution performance. In this work, we mainly focus on the former but offer a unified perspective on robustness in all of its facets.

Vulnerability of DNNs to crafted adversarial attacks (AA), diagnosed in Biggio et al. (2013), Goodfellow et al. (2015), relates to the ability of an attacker to manipulate network's outputs by changing the input images only slightly – often in ways imperceptible to a human eye. As such, AA are of key importance for security-sensitive applications and an active field of research. Most works so far have focused on attacks under *pointwise $l_p$-bounded* image distortions but a growing stream of research, pioneered by Staib and Jegelka (2017) and Sinha et al. (2018), frames the problem using Wasserstein distributionally robust optimization (DRO). We offer novel contributions to this literature.

Our key contributions can be summarized as follows. **1)** We propose a unified approach to adversarial attacks and training based on sensitivity analysis for Wasserstein DRO. We believe this approach, leveraging results from Bartl et al. (2021), is better suited for gradient-based optimization methods than duality approach adopted in most of the works to date. We further link the adversarial accuracy to the adversarial loss, and investigate the out-of-sample performance. **2)** We derive a general adversarial attack method. As a special case, this recovers the classical FGSM attack lending it a further theoretical underpinning. However, our method also allows one to carry out attacks under a *distributional threat model* which, we believe, has not been done before. We also propose a rectified DLR loss suitable for the distributional attacks. **3)** We develop asymptotically certified bounds on adversarial accuracy, applicable to a general threat, including the classical pointwise perturbations. The bounds are first-order accurate and much faster to compute than, e.g., the AutoAttack (Croce and Hein, 2020) benchmark. The performance of our methods is documented using CIFAR-10 (Krizhevsky, 2009), CIFAR-100 (Krizhevsky, 2009), ImageNet (Deng et al., 2009) datasets and neural networks from RobustBench (Croce et al., 2021).

## 2 Related Work

**Adversarial Attack (AA).** Original research focused on the *pointwise $l_p$-bounded* image distortion. Numerous attack methods under this threat model have been proposed in the literature, including Fast Gradient Sign Method (FGSM) (Goodfellow et al., 2015), Projected Gradient Descent (PGD) (Madry et al., 2018), CW attack (Carlini and Wagner, 2017), etc. In these white-box attacks, the attacker has full knowledge of the neural network. There are also black-box attacks, such as Zeroth Order Optimization (ZOO) (Chen et al., 2017), Boundary Attack (Brendel et al., 2018), and Query-limited Attack (Ilyas et al., 2018). AutoAttack (Croce and Hein, 2020), an ensemble of white-box and black-box attacks, provides a useful benchmark for *pointwise $l_p$-robustness* of neural networks. Notably Hua et al. (2022) considered AA with $l_p$ distance replaced by a proxy for human eye evaluation.

**Adversarial Defense.** Early works on data augmentation (Goodfellow et al., 2015, Madry et al., 2018, Tramèr et al., 2018) make use of strong adversarial attacks to augment the training data with adversarial examples; more recent works (Gowal et al., 2021, Xing et al., 2022, Wang et al., 2023) focus on adding randomness to training data through generative models such as GANs and diffusion models. Zhang et al. (2019) consider the trade-off between robustness and accuracy of a neural network via TRADES, a regularized loss. Analogous research includes MART (Wang et al., 2020) and SCORE (Pang et al., 2022). Other loss regularization methods such as adversarial distributional training (Dong et al., 2020) and adversarial weight perturbation (Wu et al., 2020) have been shown to smooth the loss landscape and improve the robustness. In addition, various training techniques can be overlaid to improve robustness, including droup-out layers, early stopping and parameter fine-tuning Sehwag et al. (2020). The closest to our setting are Sinha et al. (2018), García Trillos and García Trillos (2022) which employ Wasserstein penalization and constraint respectively. However, so far, even the papers which used distributional threat models to motivate DRO-based training methods actually used classical pointwise attacks to evaluate robustness of their trained DNNs. This highlights the novelty of our *distributional threat attack*.

**Robust Performance Bounds.** Each AA method gives a particular upper bound on the adversarial accuracy of the network. In contrast, research on *certified robustness* aims at classifying images which are robust to all possible attacks allowed in the threat model and thus providing an attack-agnostic lower bound on the classification accuracy. To verify robustness of images, deterministic methods using off-the-shelf solvers (Tjeng et al., 2019), relaxed linear programming (Wong and Kolter, 2018,

Weng et al., 2018a) or semi-definite programming (Raghunathan et al., 2018, Dathathri et al., 2020) have been applied. Hein and Andriushchenko (2017), Weng et al. (2018b) derive Lipschitz-based metrics to characterize the maximum distortion an image can uphold; Cohen et al. (2019) constructs a certifiable classifier by adding smooth noise to the original classifier; see Li et al. (2023) for a review.

**Distributionally Robust Optimization (DRO).** Mathematically, it is formulated as a min-max problem

$$\inf_{\theta \in \Theta} \sup_{Q \in \mathscr{P}} \mathbf{E}_Q[f_\theta(Z)], \tag{1}$$

where we minimize the worst-case loss over all possible distributions $Q \in \mathscr{P}$. In financial economics, such criteria appear in the context of multi-prior preferences, see (Gilboa and Schmeidler, 1989, Föllmer and Weber, 2015). We refer to (Rahimian and Mehrotra, 2019) for a survey of the DRO.

We focus on the Wasserstein ambiguity set $\mathscr{P} = B_\delta(P)$, which is a ball centered at the reference distribution $P$ with radius $\delta$ under the Wasserstein distance. We refer to Gao and Kleywegt (2022) for a discussion of many advantages of this distance. In particular, measures close to each other can have different supports which is key in capturing data perturbations, see Sinha et al. (2018). Staib and Jegelka (2017) interpreted *pointwise* adversarial training as a special case of Wasserstein DRO (W-DRO). Volpi et al. (2018) utilized W-DRO to improve network performance on unseen data distributions. More recently, Bui et al. (2022) unified various classical adversarial training methods, such as PGD-AT, TRADES, and MART, under the W-DRO framework.

W-DRO, while compelling theoretically, is often numerically intractable. In the literature, two lines of research have been proposed to tackle this problem. The duality approach rewrites (1), changing the sup to a univariate inf featuring a transform of $f_\theta$. We refer to Mohajerin Esfahani and Kuhn (2018) for the data-driven case, Blanchet and Murthy (2019), Bartl et al. (2020), Gao and Kleywegt (2022) for general probability measures and Huang et al. (2022) for a further application with coresets. The second approach, which we adopt here, considers the first order approximation to the original DRO problem. This can be seen as computing the sensitivity of the value function with respect to the model uncertainty as derived in Bartl et al. (2021), see also Lam (2016), García Trillos and García Trillos (2022) for analogous results in different setups.

## 3 Preliminaries

**Image Classification Task.** An image is interpreted as a tuple $(x, y)$ where the feature vector $x \in \mathcal{X}$ encodes the graphic information and $y \in \mathcal{Y} = \{1, \ldots, m\}$ denotes the class, or tag, of the image. W.l.o.g., we take $\mathcal{X} = [0, 1]^n$. A distribution of labelled images corresponds to a probability measure $P$ on $\mathcal{X} \times \mathcal{Y}$. We are given the training set $\mathcal{D}_{tr}$ and the test set $\mathcal{D}_{tt}$, subsets of $\mathcal{X} \times \mathcal{Y}$, i.i.d. sampled from $P$. We denote $\widehat{P}$ (resp. $\breve{P}$) the empirical measure of points in the training set (resp. test set), i.e., $\widehat{P} = \frac{1}{|\mathcal{D}_{tr}|} \sum_{(x,y) \in \mathcal{D}_{tr}} \delta_{(x,y)}$. A neural network is a map $f_\theta : \mathcal{X} \to \mathbb{R}^m$

$$f_\theta(x) = f^l \circ \cdots \circ f^1(x), \qquad \text{where } f^i(x) = \sigma(w^i x + b^i),$$

$\sigma$ is a nonlinear activation function, and $\theta = \{w^i, b^i : 1 \leqslant i \leqslant l\}$ is the collection of parameters. We denote $S$ the set of images equipped with their labels generated by $f_\theta$, i.e.,

$$S = \left\{ (x, y) \in \mathcal{X} \times \mathcal{Y} : \arg\max_{1 \leqslant i \leqslant m} f_\theta(x)_i = \{y\} \right\}.$$

The aim of image classification is to find a network $f_\theta$ with high (clean) prediction accuracy $A := P(S) = \mathbf{E}_P[\mathbb{1}_S]$. To this end, $f_\theta$ is trained solving[2] the stochastic optimization problem

$$\inf_{\theta \in \Theta} \mathbf{E}_P[L(f_\theta(x), y)], \tag{2}$$

where $\Theta$ denotes the set of admissible parameters, and $L$ is a (piecewise) smooth loss function, e.g., cross entropy loss[3] $\text{CE} : \mathbb{R}^m \times \mathcal{Y} \to \mathbb{R}$ given by

$$\text{CE}(z, y) = -(\log \circ \, \text{softmax}(z))_y. \tag{3}$$

---

[2] In practice, $P$ is not accessible and we use $\widehat{P}$ or $\breve{P}$ instead, e.g., in (2) we replace $P$ with $\widehat{P}$ and then compute the clean accuracy as $\breve{P}(S)$. In our experiments we make it clear which dataset is used.

[3] By convention, cross entropy is a function of two probability measures. In this case, we implicitly normalize the logit $z$ by applying $\text{softmax}$, and we associate a class $y$ with the Dirac measure $\delta_y$.

**Wasserstein Distances.** Throughout, $(p, q)$ is a pair of conjugate indices, $1/p + 1/q = 1$, with $1 \leqslant p \leqslant \infty$. We consider a norm $\| \cdot \|$ on $\mathcal{X}$ and denote $\| \cdot \|_*$ its dual, $\|\tilde{x}\|_* = \sup\{\langle x, \tilde{x}\rangle : \|x\| \leqslant 1\}$. Our main interest is in $\| \cdot \| = \| \cdot \|_r$ the $l_r$-norm for which $\| \cdot \|_* = \| \cdot \|_s$, where $(r, s)$ are conjugate indices, $1 \leqslant r \leqslant \infty$. We consider adversarial attacks which perturb the image feature $x$ but not its label $y$. Accordingly, we define a pseudo distance[4] $d$ on $\mathcal{X} \times \mathcal{Y}$ as

$$d((x_1, y_1), (x_2, y_2)) = \|x_1 - x_2\| + \infty \mathbb{1}_{\{y_1 \neq y_2\}}. \tag{4}$$

We denote $\Pi(P, Q)$ the set of couplings between $(x, y)$ and $(x', y')$ whose first margin is $P$ and second margin is $Q$, and $T_\# P := P \circ T^{-1}$ denotes the pushforward measure of $P$ under a map $T$.

The $p$-Wasserstein distance, $1 \leqslant p < \infty$, between probability measures $P$ and $Q$ on $\mathcal{X} \times \mathcal{Y}$ is

$$\mathcal{W}_p(P, Q) := \inf \left\{ \mathbf{E}_\pi [d((x_1, y_1), (x_2, y_2))^p] : \pi \in \Pi(P, Q) \right\}^{1/p}. \tag{5}$$

The $\infty$-Wasserstein distance $\mathcal{W}_\infty$ is given by

$$\mathcal{W}_\infty(P, Q) := \inf\{\pi - \mathrm{ess} \, \sup d((x_1, y_1), (x_2, y_2)) : \pi \in \Pi(P, Q)\}. \tag{6}$$

We denote the $p$-Wasserstein ball centered at $P$ with radius $\delta$ by $B_\delta(P)$. We mainly consider the cases where $p, r \in \{2, \infty\}$. Intuitively, we can view $p$ as the index of image-wise flexibility and $r$ as the index of pixel-wise flexibility. Unless $p = 1$ is explicitly allowed, $p > 1$ in what follows.

# 4 Wasserstein Distributional Robustness: adversarial attacks and training

**W-DRO Formulation.** The Wasserstein DRO (W-DRO) formulation of a DNN training task is given by:

$$\inf_{\theta \in \Theta} \sup_{Q \in B_\delta(P)} \mathbf{E}_Q[L(f_\theta(x), y)], \tag{7}$$

where $B_\delta(P)$ is the $p$-Wasserstein ball centered at $P$ and $\delta$ denotes the budget of the adversarial attack. In practice, $P$ is not accessible and is replaced with $\widehat{P}$. When $p = \infty$, the above adversarial loss coincides with the pointwise adversarial loss of Madry et al. (2018) given by

$$\inf_{\theta \in \Theta} \mathbf{E}_P[\sup\{L(f_\theta(x'), y) : \|x' - x\| \leqslant \delta\}].$$

Recently, Bui et al. (2022) considered a more general criterion they called *unified distributional robustness*. It can be re-cast equivalently as an *extended* W-DRO formulation using couplings:

$$\inf_{\theta \in \Theta} \sup_{\pi \in \Pi_\delta(P, \cdot)} \mathbf{E}_\pi[J_\theta(x, y, x', y')], \tag{8}$$

where $\Pi_\delta(P, \cdot)$ is the set of couplings between $(x, y)$ and $(x', y')$ whose first margin is $P$ and the second margin is within a Wasserstein $\delta$-ball centered at $P$. This formulation was motivated by the observation that for $p = \infty$, taking $J_\theta(x, y, x', y') = L(f_\theta(x), y) + \beta L(f_\theta(x), f_\theta(x'))$, it retrieves the TRADES loss of (Zhang et al., 2019) given by

$$\inf_{\theta \in \Theta} \mathbf{E}_P\Big[L(f_\theta(x), y) + \beta \sup_{x' : \|x - x'\| \leqslant \delta} L(f_\theta(x), f_\theta(x'))\Big].$$

**W-DRO Sensitivity.** In practice, training using (7), let alone (8), is computationally infeasible. To back propagate $\theta$ it is essential to understand the inner maximization problem denoted by

$$V(\delta) = \sup_{Q \in B_\delta(P)} \mathbf{E}_Q[J_\theta(x, y)],$$

where we write $J_\theta(x, y) = L(f_\theta(x), y)$. One can view the adversarial loss $V(\delta)$ as a certain regularization of the vanilla loss. Though we are not able to compute the exact value of $V(\delta)$ for neural networks with sufficient expressivity, DRO sensitivity analysis results allow us to derive a numerical approximation to $V(\delta)$ and further apply gradient-based optimization methods. This is the main novelty of our approach — previous works considering a W-DRO formulation mostly relied on duality results in the spirit of Blanchet and Murthy (2019) to rewrite (7).

---

[4]Our results can be adapted to regression tasks where the class label $y$ is continuous and sensitive to the perturbation. In such a setting a different $d$ would be appropriate.

**Assumption 4.1.** We assume the map $(x, y) \mapsto J_\theta(x, y)$ is L-Lipschitz under $d$, i.e.,

$$|J_\theta(x_1, y_1) - J_\theta(x_2, y_2)| \leqslant \mathsf{L}d((x_1, y_1), (x_2, y_2)).$$

The above assumption is weaker than the L-smoothness assumption encountered in the literature which requires Lipschitz continuity of gradients of $J_\theta$, see for example Sinha et al. (2018)[Assumption B] and Volpi et al. (2018)[Assumptions 1 & 2]. We also remark that our assumption holds for any continuously differentiable $J_\theta$ as the feature space $\mathcal{X} = [0, 1]^n$ is compact.

The following result follows readily from (Bartl et al., 2021, Theorem 2.2) and its proof.

**Theorem 4.1.** *Under Assumption 4.1, the following first order approximations hold:*

*(i)* $V(\delta) = V(0) + \delta\Upsilon + o(\delta)$, *where*

$$\Upsilon = \left( \mathbf{E}_P \|\nabla_x J_\theta(x, y)\|_*^q \right)^{1/q}.$$

*(ii)* $V(\delta) = \mathbf{E}_{Q_\delta}[J_\theta(x, y)] + o(\delta)$, *where*

$$Q_\delta = \left[ (x, y) \mapsto \left( x + \delta h(\nabla_x J_\theta(x, y)) \|\Upsilon^{-1} \nabla_x J_\theta(x, y)\|_*^{q-1}, y \right) \right]_\# P,$$

*and $h$ is uniquely determined by $\langle h(x), x \rangle = \|x\|_*$.*

The above holds for any probability measure, in particular with $P$ replaced consistently by an empirical measure $\widehat{P}$ or $\check{P}$. In Figure 1, we illustrate the performance of our first order approximation of the adversarial loss on CIFAR-10 (Krizhevsky, 2009) under different threat models.

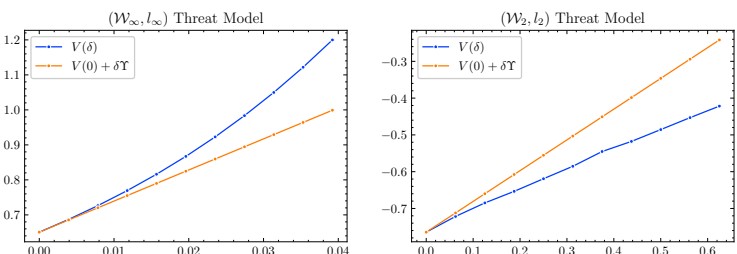

Figure 1: Performance of the first order approximation for the W-DRO value derived in Theorem 4.1. Left: WideResNet-28-10 (Gowal et al., 2020) under CE loss (3) and $(\mathcal{W}_\infty, l_\infty)$ threat model with $\delta = 1/255, \ldots, 10/255$. Right: WideResNet-28-10 (Wang et al., 2023) under ReDLR loss (10) and $(\mathcal{W}_2, l_2)$ threat models with $\delta = 1/16, \ldots, 10/16$.

**WD-Adversarial Accuracy.** We consider an attacker with perfect knowledge of the network $f_\theta$ and the data distribution $P$, aiming to minimize the prediction accuracy of $f_\theta$ under an admissible attack. Complementing the W-DRO training formulation, Staib and Jegelka (2017), Sinha et al. (2018) proposed *Wasserstein distributional threat models* under which an attack is admissible if the resulting attacked distribution $Q$ stays in the $p$-Wasserstein ball $B_\delta(P)$, where $\delta$ is the attack budget, i.e., the tolerance for distributional image distortion. We define the adversarial accuracy as:

$$A_\delta := \inf_{Q \in B_\delta(P)} Q(S) = \inf_{Q \in B_\delta(P)} \mathbf{E}_Q[\mathbb{1}_S]. \tag{9}$$

Note that $A_\delta$ is decreasing in $\delta$ with $A_0 = A$, the clean accuracy. For $p = \infty$, the Wasserstein distance essentially degenerates to the uniform distance between images and hence the proposed threat model coincides with the popular *pointwise* threat model. For $1 \leqslant p < \infty$, the *distributional* threat model is strictly stronger than the *pointwise* one, as observed in Staib and Jegelka (2017, Prop. 3.1). Intuitively, it is because the attacker has a greater flexibility and can perturb images close to the decision boundary only slightly while spending more of the attack budget on images farther away from the boundary. The threat is also closely related to out-of-distribution generalization, see Shen et al. (2021) for a survey.

**WD-Adversarial Attack.** We propose *Wasserstein distributionally adversarial attack* methods. As mentioned above, to date, even the papers which used distributional threat models to motivate DRO-based training methods then used classical pointwise attacks to evaluate robustness of their trained DNNs. Our contribution is novel and enabled by the explicit first-order expression for the distributional attack in Theorem 4.1(ii), which is not accessible using duality methods.

We recall the Difference of Logits Ratio (DLR) loss of Croce and Hein (2020). If we write $z = (z_1, \ldots, z_m) = f_\theta(x)$ for the output of a neural network, and $z_{(1)} \geqslant \cdots \geqslant z_{(m)}$ are the order statistics of $z$, then DLR loss is given by

$$\mathrm{DLR}(z, y) = \begin{cases} -\dfrac{z_y - z_{(2)}}{z_{(1)} - z_{(3)}}, & \text{if } z_y = z_{(1)}, \\ -\dfrac{z_y - z_{(1)}}{z_{(1)} - z_{(3)}}, & \text{else.} \end{cases}$$

The combination of CE loss and DLR loss has been widely shown as an effective empirical attack for *pointwise* threat models. However, under *distributional* threat models, intuitively, an effective attack should perturb more aggressively images classified far from the decision boundary and leave the misclassified images unchanged. Consequently, neither CE loss nor DLR loss are appropriate — this intuition is confirmed in our numerical experiments, see Table 1 for details. To rectify this, we propose ReDLR (Rectified DLR) loss:

$$\mathrm{ReDLR}(z, y) = -(\mathrm{DLR})^-(z, y) = \begin{cases} -\dfrac{z_y - z_{(2)}}{z_{(1)} - z_{(3)}}, & \text{if } z_y = z_{(1)}, \\ 0, & \text{else.} \end{cases} \tag{10}$$

Its key property is to leave unaffected those images that are already misclassified. Our experiments show it performs superior to CE or DLR.

An attack is performed using the test data set. For a given loss function, our proposed attack is:

$$x^{t+1} = \mathrm{proj}_\delta\big(x^t + \alpha h(\nabla_x J_\theta(x^t, y)) \|\breve{\Upsilon}^{-1} \nabla_x J_\theta(x^t, y)\|_*^{q-1}\big), \tag{11}$$

where $\alpha$ is the step size and $\mathrm{proj}_\delta$ is a projection which ensures the empirical measure $\breve{P}^{t+1} := \frac{1}{|\mathcal{D}_{tt}|} \sum_{(x,y) \in \mathcal{D}_{tt}} \delta_{(x^{t+1}, y)}$ stays inside the Wasserstein ball $B_\delta(\breve{P})$. In the case $p = r = \infty$, one can verify $h(x) = \mathrm{sgn}(x)$ and write (11) as

$$x^{t+1} = \mathrm{proj}_\delta\big(x^t + \alpha \, \mathrm{sgn}(\nabla_x J_\theta(x^t, y))\big).$$

This gives exactly Fast Gradient Sign Method (single step) and Projected Gradient Descent (multi-step) proposed in Goodfellow et al. (2015), Madry et al. (2018) and we adopt the same labels for our more general algorithms.[5] A pseudocode for the above attack is summarized in Appendix C.

Finally, note that Theorem 4.1 offers computationally tractable approximations to the *W-DRO adversarial training* objectives (7) and (8). In Appendix D we propose two possible training methods but do not evaluate their performance and otherwise leave this topic to future research.

## 5 Performance Bounds

Understanding how a DNN classifier will perform outside the training data set is of key importance. We leverage the DRO sensitivity results now to obtain a lower bound on $A_\delta$. We then use results on convergence of empirical measures in Fournier and Guillin (2015) to translate our lower bound into guarantees on out-of-sample performance.

**Bounds on Adversarial Accuracy.** We propose the following metric of robustness:

$$\mathcal{R}_\delta := \frac{A_\delta}{A} \in [0, 1].$$

Previous works mostly focus on the maximum distortion a neural network can withhold to retain certain adversarial performance, see Hein and Andriushchenko (2017), Weng et al. (2018b) for

---

[5]To stress the Wasserstein attack and the particular loss function we may write, e.g., *W-PGD-ReDLR*.

local robustness and Bastani et al. (2016) for global robustness. However, there is no immediate connection between such a maximum distortion and the adversarial accuracy, especially in face of a distributionally adversarial attack. In contrast, since $A = A_0$ is known, computing $\mathcal{R}_\delta$ is equivalent to computing $A_\delta$. We choose to focus on the relative loss of accuracy as it provides a convenient normalization: $0 \leqslant \mathcal{R}_\delta \leqslant 1$. $\mathcal{R}_\delta = 1$ corresponds to a very robust architecture which performs as well under attacks as it does on clean test data, while $\mathcal{R}_\delta = 0$ corresponds to an architecture which loses all of its predictive power under an adversarial attack. Together the couple $(A, A_\delta)$ thus summarizes the performance of a given classifier. However, computing $A_\delta$ is difficult and time-consuming. Below, we develop a simple and efficient method to calculate theoretical guaranteed bounds on $\mathcal{R}$ and thus also on $A_\delta$.

**Assumption 5.1.** We assume that for any $Q \in B_\delta(P)$

   (i) $0 < Q(S) < 1$.

   (ii) $\mathcal{W}_p(Q(\cdot|S), P(\cdot|S)) + \mathcal{W}_p(Q(\cdot|S^c), P(\cdot|S^c)) = o(\delta)$, where $S^c = (\mathcal{X} \times \mathcal{Y}) \backslash S$ and the conditional distribution is given by $Q(E|S) = Q(E \cap S)/Q(S)$.

The first condition stipulates non-degeneracy: the classifier does not perform perfectly but retains some accuracy under attacks. The second condition says the classes are well-separated: for $\delta$ small enough an admissible attack can rarely succeed.

We write the adversarial loss condition on the correctly classified images and misclassified images as

$$C(\delta) = \sup_{Q \in B_\delta(P)} \mathbf{E}_Q[J_\theta(x, y)|S] \quad \text{and} \quad W(\delta) = \sup_{Q \in B_\delta(P)} \mathbf{E}_Q[J_\theta(x, y)|S^c].$$

We note that an upper bound on $\mathcal{R}_\delta$ is given by any adversarial attack. In particular,

$$\mathcal{R}_\delta \leqslant \mathcal{R}_\delta^u := Q_\delta(S)/A. \tag{12}$$

**Theorem 5.1.** *Under Assumptions 4.1 and 5.1, we have an asymptotic lower bound as $\delta \to 0$*

$$\mathcal{R}_\delta \geqslant \frac{W(0) - V(\delta)}{W(0) - V(0)} + o(\delta) = \widetilde{\mathcal{R}}_\delta^l + o(\delta) = \overline{\mathcal{R}}_\delta^l + o(\delta), \tag{13}$$

*where the first order approximations are given by*

$$\widetilde{\mathcal{R}}_\delta^l = \frac{W(0) - \mathbf{E}_{Q_\delta}[J_\theta(x, y)]}{W(0) - V(0)} \quad \text{and} \quad \overline{\mathcal{R}}_\delta^l = \frac{W(0) - V(0) - \delta\Upsilon}{W(0) - V(0)}. \tag{14}$$

The equality between the lower bound and the two first-order approximations $\widetilde{\mathcal{R}}_\delta^l$ and $\overline{\mathcal{R}}_\delta^l$ follows from Theorem 4.1. Consequently, $\mathcal{R}_\delta^l := \min\{\widetilde{\mathcal{R}}_\delta^l, \overline{\mathcal{R}}_\delta^l\}$ allows us to estimate the model robustness without performing any sophisticated adversarial attack. Our experiments, detailed below, show the bound is reliable for small $\delta$ and is orders of magnitude faster to compute than $\mathcal{R}_\delta$ even in the classical case of pointwise attacks. The proof is reported in Appendix A. Its key ingredient is the following tower-like property.

**Proposition 5.2.** *Under Assumptions 4.1 and 5.1, we have*

$$V(\delta) = \sup_{Q \in B_\delta(P)} \mathbf{E}_Q[C(\delta)\mathbb{1}_S + W(\delta)\mathbb{1}_{S^c}] + o(\delta).$$

**Bounds on Out-of-Sample Performance.** Our results on distributionally adversarial robustness translate into bounds for performance of the trained DNN on unseen data. We rely on the results of Fournier and Guillin (2015) and refer to Lee and Raginsky (2018) for analogous applications to finite sample guarantees and to Gao (2022) for further results and discussion.

We fix $1 < p < n/2$ and let $N = |\mathcal{D}_{tr}|$, $M = |\mathcal{D}_{tt}|$. If sampling of data from $P$ is described on a probability space $(\Omega, \mathcal{F}, \mathbb{P})$ then $\widehat{P}$ is a random measure on this space and, by ergodic theorem, $\mathbb{P}$-a.s., it converges weakly to $P$ as $N \to \infty$. In fact, $\mathcal{W}_p(\widehat{P}, P)$ converges to zero $\mathbb{P}$-a.s. Crucially the rates of convergence were obtained in Dereich et al. (2013), Fournier and Guillin (2015) and yield

$$\mathbb{E}[\mathcal{W}_p(\widehat{P}, P)] \leqslant KN^{-\frac{1}{n}} \quad \text{and} \quad \mathbb{P}(\mathcal{W}_p(\widehat{P}, P) \geqslant \varepsilon) \leqslant K \exp(-KN\varepsilon^n), \tag{15}$$

where $K$ is a constant depending on $p$ and $n$ which can be computed explicitly, see for example Guo and Obłój (2019, Appendix). This, with triangle inequality and Theorem 5.1, gives

**Corollary 5.3.** *Under Assumptions 4.1 and 5.1 on measure $\widehat{P}$, with probability at least $1 - 2K \exp(-K\varepsilon^n \min\{M, N\})$ it holds that*

$$\check{A} = \check{P}(S) \geqslant \widehat{A}\widehat{\mathcal{R}}_{2\varepsilon}^l + o(\varepsilon).$$

Next results provide a finer statistical guarantee on the out-of-sample performance for robust (W-DRO) training. Its proof is reported in Appendix B.

**Theorem 5.4.** *Under Assumption 4.1, with probability at least $1 - K \exp(-KN\varepsilon^n)$ we have*

$$V(\delta) \leqslant \widehat{V}(\delta) + \varepsilon \sup_{Q \in B_\delta^\star(\widehat{P})} \left( \mathbf{E}_Q \|\nabla_x J_\theta(x, y)\|_s^q \right)^{1/q} + o(\varepsilon) \leqslant \widehat{V}(\delta) + \mathsf{L}\varepsilon$$

*where $B_\delta^\star(\widehat{P}) = \arg\max_{Q \in B_\delta(\widehat{P})} \mathbf{E}_Q[J_\theta(x, y)]$ and constant $K$ only depends on $p$ and $n$.*

Our lower bound estimate in Theorem 5.1 can be restated as

$$\Delta\widehat{A}_\delta := \widehat{A} - \widehat{A}_\delta \leqslant \frac{\widehat{V}(\delta) - \widehat{V}(0)}{\widehat{W}(0) - \widehat{C}(0)} + o(\delta).$$

We now use Theorem 5.4 to bound $\Delta A(\delta)$, the shortfall of the adversarial accuracy under $P$, using quantities evaluated under $\widehat{P}$.

**Corollary 5.5.** *Under Assumptions 4.1 and 5.1, with probability at least $1 - K \exp(-KN\delta^n)$ it holds that*

$$\Delta A_\delta(P) \leqslant \frac{\widehat{V}(\delta) - \widehat{V}(0)}{\widehat{W}(0) - \widehat{C}(0)} + \frac{2\mathsf{L}\delta}{\widehat{W}(0) - \widehat{C}(0)} + o(\delta).$$

We remark that the above results are easily extended to the out-of-sample performance on the test set, via the triangle inequality $\mathcal{W}_p(\widehat{P}, \check{P}) \leqslant \mathcal{W}_p(\widehat{P}, P) + \mathcal{W}_p(P, \check{P})$. By using complexity measures such as entropy integral (Lee and Raginsky, 2018), Rademacher complexity (Gao, 2022, Gao et al., 2022) a further analysis can be undertaken for

$$\inf_{\theta \in \Theta} \sup_{Q \in B_\delta(P)} \mathbf{E}_Q[J_\theta(x, y)] \quad \text{and} \quad \inf_{\theta \in \Theta} \sup_{Q \in B_\delta(\widehat{P})} \mathbf{E}_Q[J_\theta(x, y)]. \tag{16}$$

In particular, a dimension-free estimate of out-of-sample performance is obtained in (Gao, 2022) under a Lipschitz framework with light-tail reference measures.

## 6   Numerical Experiments

**Experimental Setting.**   We conduct experiments on a high performance computing server equipped with 49 GPU nodes. The algorithms are implemented in Python. Experiments are conducted on CIFAR-10, CIFAR-100, and ImageNet datasets. Numerical results are consistent across different datasets, and we present the results on CIFAR-10 in body paragraph only for the sake of brevity. Results on CIFAR-100 and ImageNet are reported in Appendix F.

CIFAR-10 (Krizhevsky, 2009) comprises 60,000 color images across 10 mutually exclusive classes, with 6,000 images per class. Each image contains $32 \times 32$ pixels in 3 color channels. We normalize the input feature as a vector $x \in [0, 1]^{3 \times 32 \times 32}$. The dataset is further divided into training and test sets, containing 50,000 and 10,000 images respectively. We evaluate the robustness of neural networks on the test set only.

We consider four threat models $(\mathcal{W}_p, l_r)$ with $p, r \in \{2, \infty\}$ with different range of attack budget $\delta$ depending on the relative strength of the attack. E.g., roughly speaking, if an $l_\infty$-attack modifies one third of the pixels of an image with strength 4/255, then it corresponds to an $l_2$-attack with strength 1/2. When clear from the context, we drop the $\delta$ subscript.

We take top neural networks from RobustBench (Croce et al., 2021), a lively maintained repository that records benchmark robust neural networks on CIFAR-10 against *pointwise* attacks. For *pointwise* threat models $(\mathcal{W}_\infty, l_r)$, RobustBench reports $A_\delta$ obtained using AutoAttack (Croce and Hein, 2020) for $l_\infty, \delta = 8/255$ and $l_2, \delta = 1/2$, see Appendix H. However, due to high computational

cost of AutoAttack, we apply PGD-50 based on CE and DLR losses as a substitute to obtain the reference adversarial accuracy for attacks with relatively small budgets $\delta = 2/255, 4/255$ for $l_\infty$ and $\delta = 1/8, 1/4$ for $l_2$. For *distributional* threat models $(\mathcal{W}_2, l_r)$, there is no existing benchmark attacking method. Therefore, W-PGD attack (11) based on ReDLR loss is implemented to obtain the reference adversarial accuracy $A_\delta$. All PGD attacks are run with 50 iteration steps and take between 1 and 12 hours to run on a single GPU environment. Bounds $\mathcal{R}^l, \mathcal{R}^u$ compute ca. 50 times faster.

Table 1: Comparison of adversarial accuracy of neural networks on RobustBench under different empirical attacks. Set attack budget $\delta = 8/255$ for $l_\infty$ threat models and $\delta = 1/2$ for $l_2$ threat models.

| Methods | $\mathcal{W}_\infty$ | $\mathcal{W}_2$ | | |
| | AutoAttack | W-PGD-CE | W-PGD-DLR | W-PGD-ReDLR |
| --- | --- | --- | --- | --- |
| $l_\infty$ | 57.66% | 61.32% | 79.00% | **45.46%** |
| $l_2$ | 75.78% | 74.62% | 78.69% | **61.69%** |

**Distributionally Adversarial Attack.** We report in Table 1 the average accuracy of top neural networks on RobustBench against pointwise and distributional attacks under different loss functions. The predicted drop in accuracy between a pointwise, i.e., $\infty$-W-DRO attack and a distributional 2-W-DRO attack is only realized using the ReDLR loss.

In Figure 2, we compare the adversarial accuracy of robust networks on RobustBench against *pointwise* threat models and *distributional* threat models. We notice a significant drop of the adversarial accuracy even for those neural networks robust against *pointwise* threat models.

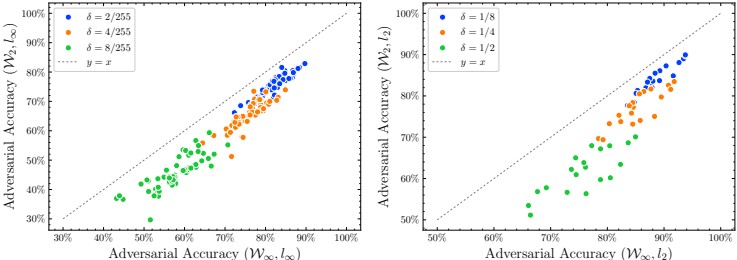

Figure 2: Shortfall of WD-adversarial accuracy with different metrics $l_\infty$ (left) and $l_2$ (right).

**Bounds on Adversarial Accuracy.** We report in Table 2 the computation time of our proposed bounds $\mathcal{R}^l_\delta = \min\{\widetilde{\mathcal{R}}^l_\delta, \overline{\mathcal{R}}^l_\delta\}$ in (14) and $\mathcal{R}^u_\delta$ in (12) with the computation time of $\mathcal{R}_\delta$ obtained from AutoAttack. Computing our proposed bounds $\mathcal{R}^l, \mathcal{R}^u$ is orders of magnitude faster than performing an attack to estimate $\mathcal{R}$. This also holds for *distributional* threat attacks.

Table 2: Computation times of $(\mathcal{W}_\infty, l_\infty), \delta = 8/255$ attack for one mini-batch of size 100, in seconds. We compute $\mathcal{R}$ by AutoAttack and average the computation time over models on RobustBench grouped by their architecture.

| | PreActResNet-18 | ResNet-18 | ResNet-50 | WRN-28-10 | WRN-34-10 | WRN-70-16 |
| --- | --- | --- | --- | --- | --- | --- |
| $\mathcal{R}$ | 197 | 175 | 271 | 401 | 456 | 2369 |
| $\mathcal{R}^l \& \mathcal{R}^u$ | 0.52 | 0.49 | 0.17 | 0.55 | 0.53 | 1.46 |

To illustrate the applications of Theorem 5.1, we plot the bounds $\mathcal{R}^l$ and $\mathcal{R}^u$ against $\mathcal{R}$ for neural networks on RobustBench. The results are plotted in Figure 3 and showcase the applicability of our bounds across different architectures.[6] Note that smaller $\delta$ values are suitable for the stronger $\mathcal{W}_2$-distributional attack. For *pointwise* threat models (top row) we compute the bounds using CE loss. For *distributional* threat models (bottom row), reference adversarial accuracy is obtained from a

---

[6]We use all 60 available networks on RobustBench (model zoo) for $l_\infty$ and all 20 available networks for $l_2$.

W-PGD-ReDLR attack and, accordingly, we use ReDLR loss to compute $\mathcal{R}^u$ and $\mathcal{R}^l$. In this case, the width of the gap between our upper and lower bounds varies significantly for different DNNs. To improve the bounds, instead of $\mathcal{R}^l$, we could estimate $V(\delta)$ and use the lower bound in (13). This offers a trade-off between computational time and accuracy which is explored further in Appendix E.

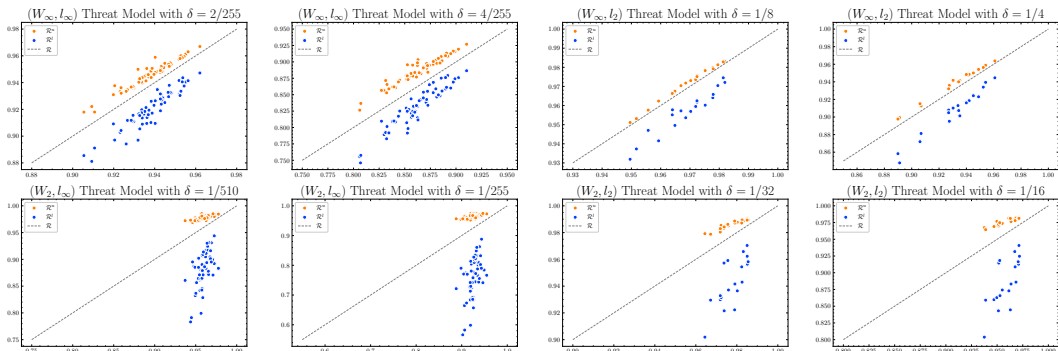

Figure 3: $\mathcal{R}^u$ & $\mathcal{R}^l$ versus $\mathcal{R}$. Top row: $\mathcal{W}_\infty$-attack with bounds computed based on CE loss across neural networks on RobustBench. Bottom row: $\mathcal{W}_2$-attack for the same sets of neural networks with bounds computed using ReDLR loss.

## 7  Limitations and Future Work

**Limitations.**  Our theoretical results are asymptotic and their validity is confined to the linear approximation regime. We believe that the empirical results we presented from all the leaderboard models on RobustBench across different attack types provide an overwhelming evidence that our results are valid and relevant for the range of attack budget $\delta$ considered in AA settings. However, as $\delta$ increases we are likely to go outside of the linear approximation regime, see Figure 1. In Appendix E we plot the results for pointwise attack with $\delta = 8/255$ where some of neural networks have a lower bound $\mathcal{R}^l$ greater than the reference $\mathcal{R}$. We do not have theoretical results to provide guarantees on the range of applicability but, in Appendix E, discuss a possible rule of thumb solution.

**Future Work.**  We believe our research opens up many avenues for future work. These include: developing stronger attacks under distributional threat models, testing the performance of the two training algorithms derived here and investigating further sensitivity-based ones, as well as analyzing the relation between the values and optimizers in (16), verifying empirical performance of our out-of-sample results, including Corollary 5.5, and extending these to out-of-distribution performance.

**Broader Impact.**  Our work contributes to the understanding of robustness of DNN classifiers. We believe it can help users in designing and testing DNN architectures. It also offers a wider viewpoint on the question of robustness and naturally links the questions of adversarial attacks, out-of-sample performance, out-of-distribution performance and Knightian uncertainty. We provide computationally efficient tools to evaluate robustness of DNNs. However, our results are asymptotic and hence valid for small attacks and we acknowledge the risk that some users may try to apply the methods outside of their applicable regimes. Finally, in principle, our work could also enhance understanding of malicious agents aiming to identify and attack vulnerable DNN-based classifiers.

## Acknowledgements

The authors are grateful to Johannes Wiesel for his most helpful comments and suggestions in the earlier stages of this project. JO gratefully acknowledges the support from St John's College, Oxford. YJ's research is supported by the EPSRC Centre for Doctoral Training in Mathematics of Random Systems: Analysis, Modelling and Simulation (EP/S023925/1). XB and GH's work, part of their research internship with JO, was supported by the Mathematical Institute and, respectively, St John's College and St Anne's College, Oxford.

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

# A    Bounds on Adversarial Accuracy

Recall that Proposition 5.2 states the following tower-like property

$$V(\delta) = \sup_{Q \in B_\delta(P)} \mathbf{E}_Q[C(\delta)\mathbb{1}_S + W(\delta)\mathbb{1}_{S^c}] = o(\delta),$$

where

$$C(\delta) = \sup_{Q \in B_\delta(P)} \mathbf{E}_Q[J_\theta(x,y)|S] \quad \text{and} \quad W(\delta) = \sup_{Q \in B_\delta(P)} \mathbf{E}_Q[J_\theta(x,y)|S^c].$$

*Proof of Proposition 5.2.* One direction follows directly from the usual tower property of conditional expectation:

$$V(\delta) = \sup_{Q \in B_\delta(P)} \mathbf{E}_Q[J(x,y)] = \sup_{Q \in B_\delta(P)} \mathbf{E}_Q[\mathbf{E}_Q[J(x,y)|\sigma(S)]]$$
$$\leqslant \sup_{Q \in B_\delta(P)} \mathbf{E}_Q[C(\delta)\mathbb{1}_S + W(\delta)\mathbb{1}_{S^c}].$$

For the other direction, note that

$$Q(E|S) = Q(E \cap S)/Q(S) \quad \text{and} \quad Q(E|S^c) = Q(E \cap S^c)/Q(S^c),$$

are well-defined for any Borel $E$ by Assumption 5.1. Take an arbitrary $\varepsilon > 0$ and $Q_c, Q_w \in B_\delta(P)$ such that

$$\mathbf{E}_{Q_c}[J(x,y)|S] \geqslant C(\delta) - \varepsilon \quad \text{and} \quad \mathbf{E}_{Q_w}[J(x,y)|S^c] \geqslant W(\delta) - \varepsilon.$$

We further take $Q_\star \in B_\delta(P)$ such that

$$\mathbf{E}_{Q_\star}[C(\delta)\mathbb{1}_S + W(\delta)\mathbb{1}_{S^c}] \geqslant \sup_{Q \in B_\delta(P)} \mathbf{E}_Q[C(\delta)\mathbb{1}_S + W(\delta)\mathbb{1}_{S^c}] - \varepsilon,$$

and write distribution $\widetilde{Q}$ given by

$$\widetilde{Q}(E) = Q_c(E|S)Q_\star(S) + Q_w(E|S^c)Q_\star(S^c).$$

These give us

$$\sup_{Q \in B_\delta(P)} \mathbf{E}_Q[C(\delta)\mathbb{1}_S + W(\delta)\mathbb{1}_{S^c}] \leqslant \mathbf{E}_{Q_\star}[C(\delta)\mathbb{1}_S + W(\delta)\mathbb{1}_{S^c}] + \varepsilon$$
$$\leqslant \mathbf{E}_{Q_\star}\big[\mathbf{E}_{Q_c}[J_\theta(x,y)|S]\mathbb{1}_S + \mathbf{E}_{Q_w}[J_\theta(x,y)|S^c]\mathbb{1}_{S^c}\big] + 3\varepsilon$$
$$= \mathbf{E}_{\widetilde{Q}}[J_\theta(x,y)] + 3\varepsilon.$$

Recall Assumption 5.1 (ii) gives for any $Q \in B_\delta(P)$

$$\mathcal{W}_p(Q(\cdot|S), P(\cdot|S)) + \mathcal{W}_p(Q(\cdot|S^c), P(\cdot|S^c)) = o(\delta).$$

Now we take $\pi_c \in \Pi(Q_\star(\cdot|S), Q_c(\cdot|S))$ such that $\mathbf{E}_{\pi_c}[d(X,Y)^p] = \mathcal{W}_p(Q_\star(\cdot|S), Q_c(\cdot|S))^p$, and similarly $\pi_w \in \Pi(Q_\star(\cdot|S^c), Q_w(\cdot|S^c))$. Then by definition of $\widetilde{Q}$, we have $\pi = Q_\star(S)\pi_c + Q_\star(S^c)\pi_w \in \Pi(Q_\star, \widetilde{Q})$. Moreover, we derive

$$\mathcal{W}_p(Q_\star, \widetilde{Q})^p \leqslant \mathbf{E}_\pi[d(X,Y)^p] = Q_\star(S)\mathbf{E}_{\pi_c}[d(X,Y)^p] + Q_\star(S^c)\mathbf{E}_{\pi_w}[d(X,Y)^p]$$
$$= Q_\star(S)\mathcal{W}_p(Q_\star(\cdot|S), Q_c(\cdot|S))^p + Q_\star(S^c)\mathcal{W}_p(Q_\star(\cdot|S^c), Q_w(\cdot|S^c))^p$$
$$= o(\delta^p),$$

which implies $\mathcal{W}_p(P, \widetilde{Q}) = \delta + o(\delta)$ by triangle inequality. Hence, by L-Lipschitzness of $J_\theta$ and (Bartl et al., 2021, Appendix Corollary 7.5) we obtain

$$\sup_{Q \in B_\delta(P)} \mathbf{E}_Q[C(\delta)\mathbb{1}_S + W(\delta)\mathbb{1}_{S^c}] \leqslant V(\delta + o(\delta)) + 3\varepsilon \leqslant V(\delta) + o(\delta) + 3\varepsilon.$$

Finally, by taking $\varepsilon \to 0$, we deduce

$$V(\delta) = \sup_{Q \in B_\delta(P)} \mathbf{E}_Q[C(\delta)\mathbb{1}_S + W(\delta)\mathbb{1}_{S^c}] + o(\delta)$$

which concludes the proof of Proposition 5.2. $\qquad\square$

Now, we present the proof of the lower bound estimate in Theorem 5.1,

$$\mathcal{R}_\delta = \frac{A_\delta}{A} \geqslant \frac{W(0) - V(0)}{W(0) - V(0)}.$$

*Proof of Theorem 5.1.* By adding and subtracting $W(\delta)\mathbb{1}_S$, Proposition 5.2 now gives

$$
\begin{aligned}
V(\delta) &= \sup_{Q \in B_\delta(P)} \mathbf{E}_Q[C(\delta)\mathbb{1}_S + W(\delta)\mathbb{1}_{S^c}] + o(\delta) \\
&= \sup_{Q \in B_\delta(P)} \mathbf{E}_Q[W(\delta) - (W_\delta - C(\delta))\mathbb{1}_S] + o(\delta) \\
&= W(\delta) - (W(\delta) - C(\delta))A_\delta + o(\delta).
\end{aligned}
$$

Naturally, we can rewrite loss $V(0)$ using the usual tower property

$$V(0) = AC(0) + (1 - A)W(0) = W(0) - (W(0) - C(0))A.$$

Together these yield

$$
\begin{aligned}
V(\delta) - V(0) &= (1 - A_\delta)(W(\delta) - W(0)) + A_\delta(C(\delta) - C(0)) + (W(0) - C(0))(A - A_\delta) + o(\delta) \\
&\geqslant (W(0) - C(0))(A - A_\delta) + o(\delta).
\end{aligned}
$$

Plugging in $C(0) = [V(0) - (1 - A)W(0)]/A$ completes the proof. $\qquad\square$

## B   Bounds on Out-of-Sample Performance

The concentration inequality in Fournier and Guillin (2015) is pivotal to derive the out-of-sample performance bounds. It characterizes how likely an empirical measure can deviate from its generating distribution. We note that recently (Larsson et al., 2023, Olea et al., 2023) obtained concentration inequalities considering other transport costs, including sliced Wasserstein distances which are of particular interest for regression tasks. Recall we denote $\widehat{P}$ as the empirical measure of training set $\mathcal{D}_{tr}$ with size $N$ and $\widecheck{P}$ as the empirical measure of test set $\mathcal{D}_{tt}$ with size $M$. We use the same $\,\widehat{}\,$, $\,\widecheck{}\,$ notations to denote quantities computed from $\widehat{P}$ and $\widecheck{P}$, respectively. We restate the concentration inequality as

$$\mathbb{P}(\mathcal{W}_p(\widehat{P}, P) \geqslant \varepsilon) \leqslant K \exp(-KN\varepsilon^n),$$

where $K$ is a constant only depending on $n$ and $p$. For convenience, $K$ might change from line to line in this section. Together with Theorem 5.1, we give an out-of-sample clean accuracy guarantee in Corollary 5.3.

*Proof of Corollary 5.3.* By triangle inequality and concentration inequality, we have

$$\mathbb{P}(\mathcal{W}_p(\widecheck{P}, \widehat{P}) \geqslant 2\varepsilon) \leqslant \mathbb{P}(\mathcal{W}_p(\widecheck{P}, P) \geqslant \varepsilon) + \mathbb{P}(\mathcal{W}_p(P, \widehat{P}) \geqslant \varepsilon) \leqslant 2K \exp(-K\varepsilon^n \min\{M, N\}).$$

With Theorem 5.1, it implies that with probability at least $1 - 2K \exp(-K\varepsilon^n \min\{M, N\})$, we have

$$\widecheck{A} \geqslant \widehat{A}_{2\varepsilon} \geqslant \widehat{A}\widehat{R}_{2\varepsilon}^l + o(\varepsilon).$$

$\qquad\square$

The following results provide a guarantee on the out-of-sample adversarial performance.

*Proof of Theorem 5.4.* Estimates in Fournier and Guillin (2015) imply that with probability at least $1 - K \exp(-KN\varepsilon^n)$, we have $B_\delta(\widehat{P}) \subseteq B_{\delta+\varepsilon}(P)$. Hence, we derive $V(\delta) \leqslant \widehat{V}(\delta + \varepsilon)$. On the other hand, since $J_\theta$ is L-Lipschitz, from (Bartl et al., 2021, Appendix Corollary 7.5) we obtain

$$\widehat{V}(\delta + \varepsilon) = \widehat{V}(\delta) + \varepsilon \sup_{Q \in B_\delta^\star(\widehat{P})} \left(\mathbf{E}_Q \|\nabla_x J_\theta(x, y)\|_*^q\right)^{1/q} + o(\varepsilon),$$

where $B_\delta^\star(\widehat{P}) = \arg\max_{Q \in B_\delta(\widehat{P})} \mathbf{E}_Q[J_\theta(x, y)]$. Combining above results, we conclude that with probability at least $1 - K \exp(-KN\varepsilon^n)$

$$V(\delta) \leqslant \widehat{V}(\delta) + \varepsilon \sup_{Q \in B_\delta^\star(\widehat{P})} \left(\mathbf{E}_Q \|\nabla_x J_\theta(x, y)\|_*^q\right)^{1/q} + o(\varepsilon) \leqslant \widehat{V}(\delta) + \mathsf{L}\varepsilon.$$

$\qquad\square$

*Proof Corollary 5.5.* By Theorem 5.1, we have

$$\Delta A_\delta(P) \leqslant \frac{V(\delta) - V(0)}{W(0) - C(0)} + o(\delta).$$

We now bound the numerator and denominator separately. By taking $\varepsilon = \delta$ in Theorem 5.4, we have with probability at least $1 - K \exp(-KN\delta^n)$, $V(\delta) \leqslant \widehat{V}(\delta) + \mathsf{L}\delta$. Similarly, we can also show that $V(0) \geqslant \widehat{V}(0) - \mathsf{L}\delta$. Hence, we have with probability at least $1 - K \exp(-KN\delta^n)$,

$$V(\delta) - V(0) \leqslant \widehat{V}(\delta) - \widehat{V}(0) + 2\mathsf{L}\delta.$$

For the denominator, notice that $\mathbb{P}(\widehat{P} \in B_\delta(P)) \geqslant 1 - K \exp(-KN \exp(\delta^n))$. By Assumption 5.1 (ii), we have with probability at least $1 - K \exp(-KN\delta^n)$,

$$\mathcal{W}_p(\widehat{P}(\cdot|S), P(\cdot|S)) + \mathcal{W}_p(\widehat{P}(\cdot|S^c), P(\cdot|S^c)) = o(\delta).$$

This implies

$$|C(0) - W(0) - \widehat{C}(0) + \widehat{W}(0)| = o(\delta).$$

Combining above results, we conclude that with probability at least $1 - K \exp(-KN\delta^n)$,

$$\Delta A_\delta(P) \leqslant \frac{\widehat{V}(\delta) - \widehat{V}(0)}{\widehat{W}(0) - \widehat{C}(0)} + \frac{2\mathsf{L}\delta}{\widehat{W}(0) - \widehat{C}(0)} + o(\delta).$$

$\square$

## C    W-PGD Algorithm

We give the details of W-PGD algorithm implemented in this paper. Recall in Section 4, we propose

$$x^{t+1} = \mathrm{proj}_\delta\big(x^t + \alpha h(\nabla_x J_\theta(x^t, y))\|\Upsilon^{-1}\nabla_x J_\theta(x^t, y)\|_s^{q-1}\big), \tag{17}$$

where we take $\|\cdot\| = \|\cdot\|_r$ and hence $h$ is given by $\langle h(x), x\rangle = \|x\|_s$. In particular, $h(x) = \mathrm{sgn}(x)$ for $s = 1$ and $h(x) = x/\|x\|_2$ for $s = 2$. The projection step $\mathrm{proj}_\delta$ is based on any off-the-shelf optimal transport solver $\mathscr{S}$ which pushes the adversarial images back into the Wasserstein ball along the geodesics. The solver $\mathscr{S}$ gives the optimal matching $T$ between the original test data $\mathcal{D}_{tt}$ and the perturbed test data $\mathcal{D}'_{tt}$. Formally, $\mathrm{proj}_\delta$ maps

$$x' \mapsto x + \delta d^{-1}(T(x) - x),$$

where $d = \left\{\frac{1}{|\mathcal{D}_{tt}|} \sum_{(x,y)\in\mathcal{D}_{tt}} \|T(x) - x\|_r^p\right\}^{1/p}$ the Wasserstein distance between $\check{P}$ and $\check{P}'$. See Algorithm 1 for pseudocodes. In numerical experiments, due to the high computational cost of the OT solver, we always couple each image with its own perturbation.

**Algorithm 1:** W-PGD Attack

**Input:** Model parameter $\theta$, attack strength $\delta$, ratio $r$, iteration step $I$, OT solver $\mathscr{S}$;
**Data:** Test set $\mathcal{D}_{tt} = \{(x,y)|(x,y) \sim P\}$ with size $M$;
**def** $\mathrm{proj}_\delta(\mathcal{D}_{tt}, \mathcal{D}'_{tt})$**:**
  $\quad T = \mathscr{S}(\mathcal{D}_{tt}, \mathcal{D}'_{tt})$;                    // Generate transport map from OT solver
  $\quad d = \left\{ \frac{1}{M} \sum_{(x,y) \in \mathcal{D}_{tt}} \|T(x) - x\|_r^p \right\}^{1/p}$;    // Calculate the Wasserstein distance
  $\quad$**for** $(x,y)$ *in* $\mathcal{D}_{tt}$ **do**
  $\quad\quad x' \leftarrow x + \delta d^{-1}(T(x) - x)$;          // Project back to the Wasserstein ball
  $\quad\quad x' \leftarrow \mathrm{clamp}(x', 0, 1)$;
  $\quad$**return** $\mathcal{D}'_{tt}$.
**def** $\mathrm{attack}(\mathcal{D}_{tt})$**:**
  $\quad \alpha \leftarrow r\delta/I$;                              // Calculate stepsize
  $\quad \mathcal{D}_{tt}^{adv} \leftarrow \mathcal{D}_{tt}$;
  $\quad$**for** $1 \leqslant i \leqslant I$ **do**
  $\quad\quad \Upsilon = \left( \frac{1}{M} \sum_{(x,y) \in \mathcal{D}_{tt}^{adv}} \|\nabla_x J_\theta(x,y)\|_s^q \right)^{1/q}$;                    // Calculate $\Upsilon$
  $\quad\quad$**for** $(x,y) \in \mathcal{D}_{tt}^{adv}$ **do**
  $\quad\quad\quad (x,y) \leftarrow \big(x + \alpha h(\nabla_x J_\theta(x,y))\|\Upsilon^{-1} \nabla_x J_\theta(x,y)\|_s^{q-1}, y\big)$;
  $\quad\quad \mathcal{D}_{tt}^{adv} = \mathrm{proj}_\delta(\mathcal{D}_{tt}, \mathcal{D}_{tt}^{adv})$;
  $\quad\quad$**return** $\mathcal{D}_{tt}^{adv}$.

# D  Wasserstein Distributionally Adversarial Training

Theorem 4.1 offers natural computationally tractable approximations to the W-DRO training objective

$$\inf_{\theta \in \Theta} \sup_{Q \in B_\delta(P)} \mathbf{E}_Q[J_\theta(x,y)]$$

and its extension

$$\inf_{\theta \in \Theta} \sup_{\pi \in \Pi_\delta(P,\cdot)} \mathbf{E}_\pi[J_\theta(x,y,x',y')].$$

First, consider a regularized optimization problem:

$$\inf_{\theta \in \Theta} \mathbf{E}_P[J_\theta(x,y) + \delta\Upsilon].$$

The extra regularization term $\delta\Upsilon$ allows us to approximate the W-DRO objective above. A similar approach has been studied in García Trillos and García Trillos (2022) in the context of Neural ODEs, and Sinha et al. (2018) considered Wasserstein distance penalization and used duality.

Training is done by replacing $P$ with $\widehat{P}$. Note that $\widehat{\Upsilon}$ is a statistics over the whole data set. In order to implement stochastic gradient descent method, an asynchronous update of the parameters is needed. We consider $\|\cdot\|_* = \|\cdot\|_s$ and, by a direct calculation, obtain

$$\nabla_\theta \widehat{\Upsilon} = \widehat{\Upsilon}^{1-q} \mathbf{E}_{\widehat{P}}[\langle \nabla_x \nabla_\theta J_\theta(x,y), \mathrm{sgn}(\nabla_x J_\theta(x,y))\rangle \|\nabla_x J_\theta(x,y)\|_s^{q-1}].$$

We calculate the term $\widehat{\Upsilon}^{1-q}$ from the whole training dataset using parameter $\theta^*$ from previous epoch; we estimate

$$\mathbf{E}_{\widehat{P}}[\langle \nabla_x \nabla_\theta J_\theta(x,y), \mathrm{sgn}(\nabla_x J_\theta(x,y))\rangle \|\nabla_x J_\theta(x,y)\|_s^{q-1}]$$

on a mini-batch and update current parameter $\theta$ after each batch. At the end of an epoch, we update $\theta^*$ to $\theta$. See Algorithm 2 for pseudocodes.

Another classical approach consists in clean training the network but on the adversarial perturbed data. To this we employ the Wasserstein FGSM attack described in Section 4 and shift training data $(x,y)$ by

$$(x,y) \mapsto \Big( \mathrm{proj}_\delta\big(x + \delta h(\nabla_x J_\theta(x,y))\|\widehat{\Upsilon}^{-1} \nabla_x J_\theta(x,y)\|_s^{q-1}\big), y\Big).$$

Similarly to the discussion above, an asynchronous update of parameters is applied, see Algorithm 3 for details. Empirical evaluation of the performance of Algorithms 2 and 3 is left for future research.

---

**Algorithm 2:** Loss Regularization

---

**Input:** Initial parameter $\theta_0$, hyperparameter $\delta$, learning rate $\eta$;
**Data:** Training set $\mathcal{D}_{tr} = \{(x,y)|(x,y) \sim P\}$ with size $N$;
$\theta^* \leftarrow \theta_0, \theta \leftarrow \theta_0$;
**repeat**

$\quad \Upsilon = \left( \frac{1}{N} \sum_{(x,y)\in\mathcal{D}_{tr}} \|\nabla_x J_{\theta*}(x,y)\|_s^q \right)^{1/q}$;          `// Calculate` $\Upsilon$ `from` $\theta^*$

$\quad$ **repeat**

$\quad\quad$ Generate a mini-batch $B$ with size $|B|$;
$\quad\quad$ `// Calculate gradient` $\nabla_\theta J_\theta(x,y)$
$\quad\quad \nabla_\theta J_\theta(x,y) = \frac{1}{|B|} \sum_{(x,y)\in B} \nabla_\theta J_\theta(x,y)$;
$\quad\quad$ `// Calculate gradient` $\nabla_\theta \Upsilon$
$\quad\quad \nabla_\theta \Upsilon = \Upsilon^{1-q} \frac{1}{|B|} \sum_{(x,y)\in B} \langle \nabla_x \nabla_\theta J_\theta(x,y), h(\nabla_x J_\theta(x,y)) \rangle \|\nabla_x J_\theta(x,y)\|_s^{q-1}$;
$\quad\quad$ `// Update` $\theta$ `by stochastic gradient descent`
$\quad\quad \theta \leftarrow \theta - \eta(\nabla_\theta J_\theta(x,y) + \delta \nabla_\theta \Upsilon)$;

$\quad$ **until** *the end of epoch;*
$\quad \theta^* \leftarrow \theta$;
**until** *the end condition.*

---

---

**Algorithm 3:** Adversarial Data Perturbation

---

**Input:** Initial parameter $\theta_0$, hyperparameter $\delta$, learning rate $\eta$;
**Data:** Training set $\mathcal{D}_{tr} = \{(x,y)|(x,y) \sim P\}$ with size $N$;
$\theta^* \leftarrow \theta_0, \theta \leftarrow \theta_0$;
**repeat**

$\quad \Upsilon = \left( \frac{1}{N} \sum_{(x,y)\in\mathcal{D}_{tr}} \|\nabla_x J_{\theta*}(x,y)\|_s^q \right)^{1/q}$;          `// Calculate` $\Upsilon$ `from` $\theta^*$

$\quad$ **repeat**

$\quad\quad$ Generate a mini-batch $B$ with size $|B|$;
$\quad\quad$ `// Do W-FGSM attack on` $B$
$\quad\quad (x,y) \leftarrow \left( \text{proj}_\delta(x + \delta h(\nabla_x J_\theta(x,y)) \|\Upsilon^{-1} \nabla_x J_\theta(x,y)\|_s^{q-1}), y \right)$;
$\quad\quad$ `// Update` $\theta$ `by stochastic gradient descent`
$\quad\quad \theta \leftarrow \theta - \eta \frac{1}{|B|} \sum_{(x,y)\in B} \nabla_\theta J_\theta(x,y)$;

$\quad$ **until** *the end of epoch;*
$\quad \theta^* \leftarrow \theta$;
**until** *the end condition.*

---

## E   Robust Performance Bounds

As pointed out in Section 7, with attack budget $\delta = 8/255$ some neural networks have lower bounds $\mathcal{R}^l$ surpassing the reference value $\mathcal{R}$ obtained from AutoAttack. It is because for most of neural networks $\delta = 8/255$ is outside the linear approximation region of the adversarial loss $V(\delta)$, and we underestimate $V(\delta)$ by using first order approximations in Theorem 4.1. In Figure 4, we plot our proposed bounds $\mathcal{R}^u$ and $\mathcal{R}^l$ against $\mathcal{R}$ for $(\mathcal{W}_\infty, l_\infty)$ threat model with $\delta = 8/255$.

In general, to compute more accurate lower bounds on the adversarial accuracy, as explained in Section 6, we can consider the first lower bound in (13). We thus introduce $\mathcal{R}^l(n)$ given by

$$\mathcal{R}^l(n) = \frac{W(0) - V(\delta, n)}{W(0) - V(0)},$$

where $V(\delta, n)$ is the approximated adversarial loss computed by a W-PDG-($n$) attack. In Figures 5 and 6, we include plots for different bounds of $\mathcal{R}$ under $\mathcal{W}_2$ threat models which illustrate the changing performance of the lower bound in Theorem 5.1 as $V(\delta)$ is computed to an increasing accuracy. We achieve this performing a W-PDG-($n$) attack, where $n = 5, 50$. An $n = 50$ attack takes

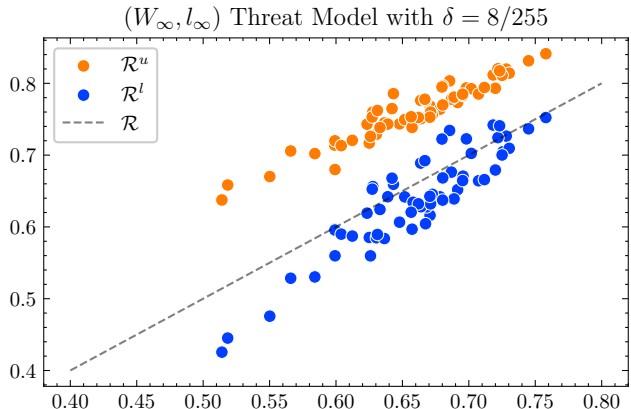

Figure 4: $\mathcal{R}^u \& \mathcal{R}^l$ versus $\mathcal{R}$. Bounds are computed based on CE loss across neural networks on RobustBench under a $(\mathcal{W}_\infty, l_\infty)$ threat model with budget $\delta = 8/255$. At this $\delta$ linear approximation may be inefficient as seen from the blue dots crossing the diagonal.

10 times more computational time than the $n = 5$ attack and the latter is 5 times more computational costly than the one-step bound $\mathcal{R}^l$. The plots thus illustrate a trade-off between computational time and accuracy of the proposed lower bound. For clarity, we also point out that Figure 5 has a different scaling from the other plots.

In practice, we would compare our one-step-attack bound $\mathcal{R}^l$ with the bound obtained by iterating the attack several steps: we propose W-PGD(5) in our paper and report $\mathcal{R}^l(5)$. If the difference between the two is small it indicates linear approximation is working very well. If the difference is significant, we would use $\mathcal{R}^l(5)$ and maybe compare it against $\mathcal{R}^l(10)$. If we observe that $V(\delta)$ is convex - as for the CE loss under $(\mathcal{W}_\infty, l_\infty)$ attack - the lower bound should decrease. In this case, the one-step bound may in fact not be a lower bound for too large as shown in Figure 4. If we observe $V(\delta)$ is concave - as for the ReDLR loss and our $\mathcal{W}_2$ attacks - the one-step lower bound might be too low and will increase with additional PGD steps. This is visible in Figures 5 and 6.

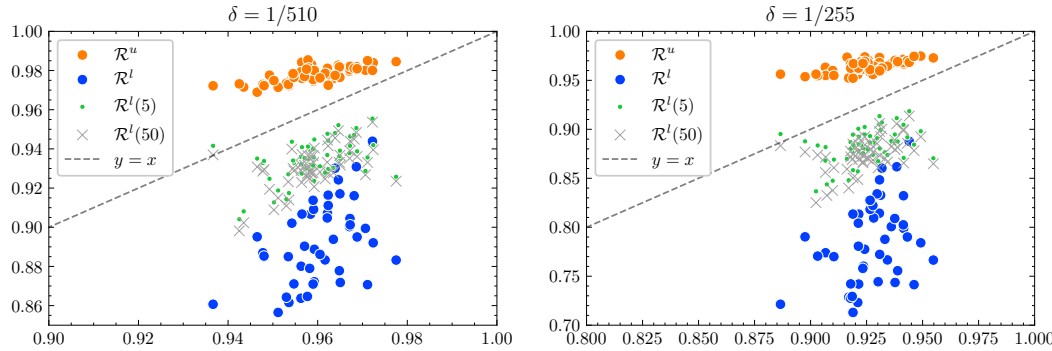

Figure 5: Comparison of lower bounds computed from different W-PDG-$(n)$ attacks, where $n = 5, 50$. Bounds are computed based on CE loss across neural networks on RobustBench under $(\mathcal{W}_2, l_\infty)$ threat models with budget $\delta = 1/510, 1/255$.

## F  Experiments on Other Datasets

We illustrate our theoretical results on CIFAR-100 (Krizhevsky, 2009) and ImageNet (Deng et al., 2009) datasets. We analyze all the networks included on RobustBench leaderboards. Only $l_\infty$ norm on images is considered as RobustBench does not provide any models for $l_2$ for these datasets. Similar to CIFAR-10, CIFAR-100 is a low resolution dataset but with 100 classes. Each class contains 500 training images and 100 testing images, and each image has $32 \times 32$ pixels in 3 color channels. For

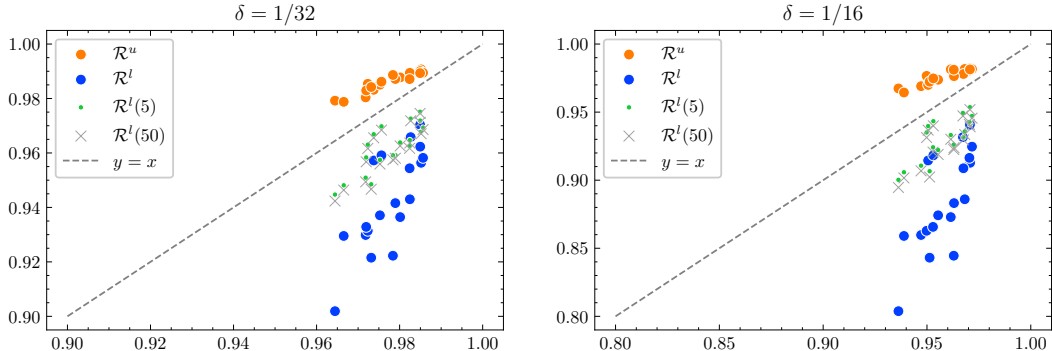

Figure 6: Comparison of lower bounds computed from different W-PDG-($n$) attacks, where $n = 5, 50$. Bounds are computed based on ReDLR loss across neural networks on RobustBench under $(\mathcal{W}_2, l_2)$ threat models with budget $\delta = 1/32, 1/16$.

ImageNet dataset, we use a subset of 5000 images selected by RobustBench. The size of image may vary among different preprocessing. A typical size is $224 \times 224$ pixels in 3 color channels. Numerical results mirror those for CIFAR-10 in the body paragraph.

Before the discussion on the numerical results, we remark that the computation time of our proposed methods do not suffer from the curse of dimensionality. Despite the fact that complexity of Wasserstein distance in higher dim can cause troubles, this does not happen for our methods due to our formulae for the first order sensitivity. This means that the approximation to the worst case Wasserstein adversarial attack is explicit and only requires us to compute the gradient $\nabla_x J_\theta(x, y)$. In consequence, the size of the image only affects the first layer of the DNN and the number of classes affects the last layer of the DNN. The computational time is thus mostly determined by the (hidden) architecture of the DNN model we consider.

In Figure 7, we compare the adversarial accuracy under *pointwise* threat models and *distributional* threat models. For $\mathcal{W}_\infty$ threat model, we apply a combination PGD-50 attack based on CE, DLR losses to obtain the reference adversarial accuracy; for $\mathcal{W}_2$ threat model, we apply a 50-step W-PGD-ReDLR to obtain the reference WD-adversarial accuracy. Similar to CIFAR-10, we observe a drop of adversarial accuracy under distributional threat models on CIFAR-100 and ImageNet.

In Figure 8, we plot the bounds $\mathcal{R}^l$ and $\mathcal{R}^u$ against $\mathcal{R}$ on CIFAR-100 and ImageNet. We compute bounds under $\mathcal{W}_\infty$-attack based on CE loss across neural networks on RobustBench; $\mathcal{W}_2$-attack for the same sets of neural networks with bounds computed using ReDLR loss, where $\mathcal{R}^l(5)$ denotes the lower bound (eq. (13)) computed from a 5-step W-PGD attack. For pointwise threat models, our upper and lower bounds sandwich the true (AutoAttack) accuracy ratio very well at a fraction of the computational cost. For distributional threat models, our bounds work but the tightness of the bound is model-dependent. We display also the improved bound obtained with 5 steps of W-PGD attack which leads to a marked improvement in the lower bound's performance. This is akin to Figure 5 in Appendix E.

## G    Comparison to Existing Attack Methods

In this section, we compare our proposed W-PGD attack with existing attack methods. We reiterate that our method focuses on *distributional* threats under a *given* budget $\delta$. While the W-DRO formulation has been explored in numerous works, as discussed, we believe that none of them proposed an AA method specifically tailored for the distributional threat under a given attack budget.

We illustrate the main difference between our sensitivity approach and duality approach employed in Sinha et al. (2018), Volpi et al. (2018), Hua et al. (2022), etc. When using duality of W-DRO, the optimization involves a minimization over the Lagrangian multiplier $\lambda$. However, in the aforementioned works, a fixed $\lambda$ is used. While their choice of $\lambda$ may be optimal for *some* budget $\delta$, there is no universal approach to achieve an optimal attack for a *given* budget $\delta$. In contrast, our attack is an explicit first-order approximation designed to be optimal for a *given* budget $\delta$. It only involves $\nabla_x J_\theta(x, y)$ and its norms which are fast to compute using standard methods.

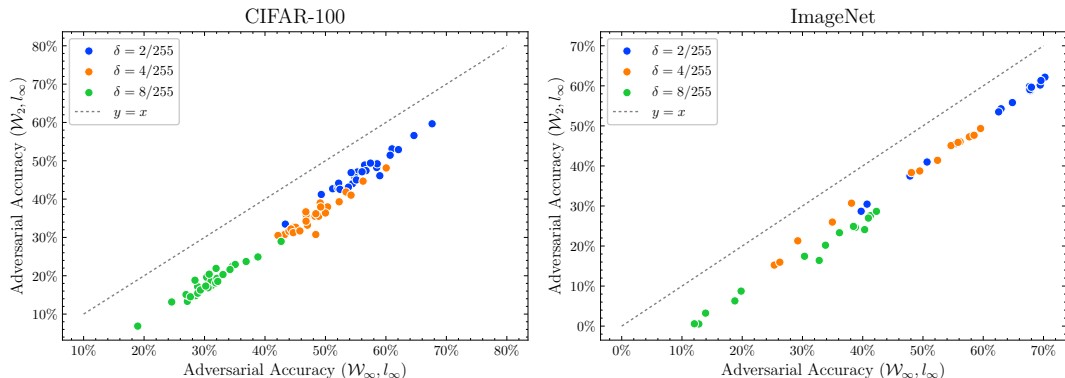

Figure 7: Shortfall of WD-adversarial accuracy on CIFAR-100 (left) and ImageNet (right).

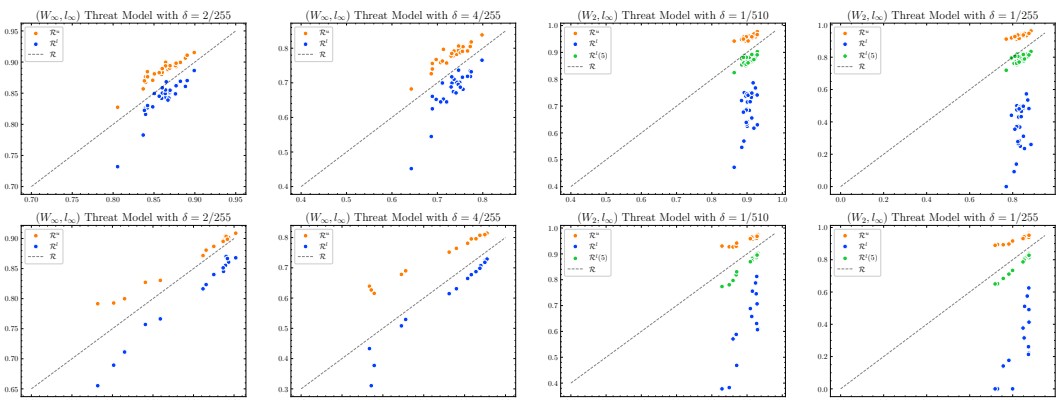

Figure 8: $\mathcal{R}^u$ & $\mathcal{R}^l$ versus $\mathcal{R}$ on CIFAR-100 (top) and ImageNet (bottom).

Despite no existing baseline for addressing distributional threat, it is possible to reverse engineer a comparison setup ex post. We employ CW attack which gives the minimum distortion needed to successfully attack an image. Due to the computational cost of CW attack, the experiment is conducted on the first 1000 correctly classified images from the ImageNet subset used by RobustBench. By calculating the square root of the mean square of the minimum distortion over this chosen dataset, we obtain a reference $(\mathcal{W}_2, l_\infty)$ budget 0.000711 to perform our distributional attacks. Employing our W-PGD(50)-ReDLR attack results in 934 images being successfully attacked out of the 1000. The CW attack, which by construction was successful on all 1000 images, took ca 4.5h to run (on a Tesla V100-SXM2-32GB-LS GPU) and the time needed to be scaled linearly with the number of images as the attack has to be performed image by image. In contrast, our attack took ca 50min to run (on a NVIDIA A100-SXM4-40GB GPU). In addition, we use mini-batches, which allows for computational speed improvements based on the available GPU memory.

## H    Additional Numerical Results

In Tables 3 and 4, we report the clean accuracy and the adversarial accuracy under different threat models for all but 4 neural networks available on RobustBench (model zoo). All networks are named after their labels on RobustBench (model zoo). We remove Standard $l_\infty$-network and Standard $l_2$-network because they are not robust to any attacks. In addition, we also remove Kang2021Stable which contains NeuralODE blocks and Ding2020MMA which has a huge gap between adversarial accuracies obtained from PGD and AutoAttack.

In Tables 5, 6, 7 and 8, we include the complete list of $\mathcal{R}$ and its bounds used in Figure 3. We write $\Delta\mathcal{R}^u = \mathcal{R}^u - \mathcal{R}$ and $\Delta\mathcal{R}^l = \mathcal{R}^l - \mathcal{R}$.

Table 3: Complete list of adversarial accuracies under $(\mathcal{W}_\infty, l_\infty)$ and $(\mathcal{W}_2, l_\infty)$ threat models.

| | | $\delta=1/8$ | | $\delta=1/4$ | | $\delta=1/8$ | |
|---|---|---|---|---|---|---|---|
| Network | Clean | $\mathcal{W}_\infty$ | $\mathcal{W}_2$ | $\mathcal{W}_\infty$ | $\mathcal{W}_2$ | $\mathcal{W}_\infty$ | $\mathcal{W}_2$ |
| Augustin2020Adversarial | 91.08 | 87.88 | 82.03 | 84.48 | 73.17 | 72.91 | 56.66 |
| Augustin2020Adversarial_34_10 | 92.23 | 89.20 | 83.70 | 85.80 | 74.06 | 76.25 | 56.33 |
| Augustin2020Adversarial_34_10_extra | 93.96 | 91.63 | 84.86 | 88.32 | 75.04 | 78.79 | 59.73 |
| Ding2020MMA | 88.02 | 83.57 | 77.69 | 78.44 | 69.67 | 66.09 | 53.47 |
| Engstrom2019Robustness | 90.83 | 86.81 | 81.99 | 82.37 | 73.77 | 69.24 | 57.77 |
| Gowal2020Uncovering | 90.89 | 87.62 | 83.19 | 84.26 | 75.83 | 74.50 | 60.94 |
| Gowal2020Uncovering_extra | 94.73 | 92.64 | 88.08 | 89.52 | 79.74 | 80.53 | 60.18 |
| Rade2021Helper_R18_ddpm | 90.57 | 88.03 | 83.57 | 84.58 | 77.26 | 76.15 | 62.74 |
| Rebuffi2021Fixing_28_10_cutmix_ddpm | 91.79 | 89.26 | 86.09 | 86.43 | 81.06 | 78.80 | 67.18 |
| Rebuffi2021Fixing_70_16_cutmix_ddpm | 92.41 | 90.36 | 87.27 | 87.69 | 81.67 | 80.42 | 67.93 |
| Rebuffi2021Fixing_70_16_cutmix_extra | 95.74 | 93.78 | 89.73 | 91.18 | 81.59 | 82.32 | 63.43 |
| Rebuffi2021Fixing_R18_cutmix_ddpm | 90.33 | 87.48 | 84.24 | 84.48 | 78.38 | 75.86 | 63.85 |
| Rice2020Overfitting | 88.68 | 85.07 | 80.62 | 80.35 | 73.28 | 67.68 | 56.83 |
| Rony2019Decoupling | 89.04 | 84.74 | 78.47 | 79.25 | 69.39 | 66.44 | 51.15 |
| Sehwag2021Proxy | 90.93 | 88.42 | 85.50 | 85.66 | 80.50 | 77.24 | 67.96 |
| Sehwag2021Proxy_R18 | 89.76 | 87.08 | 83.34 | 83.87 | 77.62 | 74.41 | 65.02 |
| Wang2023Better_WRN-28-10 | 95.16 | 93.44 | 88.99 | 90.79 | 82.58 | 83.68 | 68.65 |
| Wang2023Better_WRN-70-16 | 95.54 | 93.76 | 89.94 | 91.79 | 83.47 | 84.97 | 70.09 |
| Wu2020Adversarial | 88.51 | 85.32 | 81.33 | 82.05 | 75.31 | 73.66 | 62.21 |

Table 4: Complete list of adversarial accuracies under $(\mathcal{W}_\infty, l_\infty)$ and $(\mathcal{W}_2, l_\infty)$ threat models.

| Network | Clean | $\delta$=2/255 | | $\delta$=4/255 | | $\delta$=8/255 | |
|---|---|---|---|---|---|---|---|
| | | $\mathcal{W}_\infty$ | $\mathcal{W}_2$ | $\mathcal{W}_\infty$ | $\mathcal{W}_2$ | $\mathcal{W}_\infty$ | $\mathcal{W}_2$ |
| Addepalli2021Towards_RN18 | 80.23 | 73.85 | 68.54 | 66.70 | 58.76 | 51.06 | 42.91 |
| Addepalli2021Towards_WRN34 | 85.32 | 79.87 | 73.41 | 73.18 | 64.60 | 58.04 | 48.12 |
| Addepalli2022Efficient_RN18 | 85.71 | 79.16 | 71.75 | 71.37 | 59.59 | 52.48 | 37.84 |
| Addepalli2022Efficient_WRN_34_10 | 88.70 | 82.95 | 74.75 | 75.52 | 63.35 | 57.81 | 42.03 |
| Andriushchenko2020Understanding | 79.85 | 72.30 | 66.16 | 64.45 | 55.83 | 43.93 | 37.86 |
| Carmon2019Unlabeled | 89.69 | 84.59 | 77.02 | 77.87 | 66.59 | 59.53 | 46.32 |
| Chen2020Adversarial | 86.04 | 79.68 | 67.45 | 71.61 | 51.23 | 51.56 | 29.66 |
| Chen2020Efficient | 85.34 | 78.48 | 70.91 | 70.64 | 59.24 | 51.12 | 39.32 |
| Chen2021LTD_WRN34_10 | 85.21 | 79.78 | 72.74 | 72.97 | 61.86 | 56.94 | 41.48 |
| Chen2021LTD_WRN34_20 | 86.03 | 80.71 | 75.31 | 74.19 | 64.95 | 57.71 | 44.87 |
| Cui2020Learnable_34_10 | 88.22 | 81.95 | 71.25 | 74.20 | 57.88 | 52.86 | 40.14 |
| Cui2020Learnable_34_20 | 88.70 | 82.20 | 72.16 | 74.46 | 57.74 | 53.57 | 37.73 |
| Dai2021Parameterizing | 87.02 | 82.39 | 77.35 | 76.96 | 69.28 | 61.55 | 52.43 |
| Debenedetti2022Light_XCiT-L12 | 91.73 | 86.59 | 78.09 | 79.42 | 66.77 | 57.58 | 44.90 |
| Debenedetti2022Light_XCiT-M12 | 91.30 | 86.14 | 77.83 | 78.89 | 66.86 | 57.27 | 43.90 |
| Debenedetti2022Light_XCiT-S12 | 90.06 | 84.91 | 77.11 | 77.50 | 65.35 | 56.14 | 44.28 |
| Engstrom2019Robustness | 87.03 | 80.34 | 74.30 | 72.22 | 63.43 | 49.25 | 41.85 |
| Gowal2020Uncovering_28_10_extra | 89.48 | 84.78 | 78.08 | 78.77 | 67.93 | 62.80 | 48.51 |
| Gowal2020Uncovering_34_20 | 85.64 | 79.84 | 73.22 | 73.47 | 62.78 | 56.86 | 42.87 |
| Gowal2020Uncovering_70_16 | 85.29 | 79.81 | 72.89 | 73.47 | 62.25 | 57.20 | 43.29 |
| Gowal2020Uncovering_70_16_extra | 91.10 | 86.86 | 79.81 | 81.06 | 70.01 | 65.88 | 50.55 |
| Gowal2021Improving_28_10_ddpm_100m | 87.50 | 83.37 | 78.46 | 78.30 | 70.50 | 63.44 | 54.95 |
| Gowal2021Improving_70_16_ddpm_100m | 88.74 | 84.76 | 80.48 | 80.08 | 73.27 | 66.11 | 59.34 |
| Gowal2021Improving_R18_ddpm_100m | 87.35 | 82.08 | 76.83 | 76.22 | 68.16 | 58.63 | 51.17 |
| Hendrycks2019Using | 87.11 | 81.36 | 74.48 | 74.21 | 63.45 | 54.92 | 44.19 |
| Huang2020Self | 83.48 | 77.59 | 69.33 | 70.55 | 58.46 | 53.34 | 38.96 |
| Huang2021Exploring | 90.56 | 85.77 | 77.78 | 79.50 | 67.52 | 61.56 | 47.30 |
| Huang2021Exploring_ema | 91.23 | 86.84 | 79.28 | 80.79 | 69.02 | 62.54 | 48.46 |
| Huang2022Revisiting_WRN-A4 | 91.59 | 87.35 | 79.76 | 81.69 | 69.62 | 65.79 | 50.22 |
| Jia2022LAS-AT_34_10 | 84.98 | 79.26 | 73.00 | 72.44 | 62.88 | 56.26 | 43.92 |
| Jia2022LAS-AT_70_16 | 85.66 | 80.29 | 73.90 | 73.52 | 63.76 | 57.61 | 44.19 |
| Pang2020Boosting | 85.14 | 79.34 | 71.93 | 72.60 | 61.17 | 53.74 | 39.40 |
| Pang2022Robustness_WRN28_10 | 88.61 | 83.73 | 78.46 | 77.16 | 69.51 | 61.04 | 51.67 |
| Pang2022Robustness_WRN70_16 | 89.01 | 84.77 | 79.57 | 78.58 | 70.61 | 63.35 | 52.93 |
| Rade2021Helper_ddpm | 88.16 | 83.26 | 77.54 | 76.83 | 67.49 | 60.97 | 47.37 |
| Rade2021Helper_extra | 91.47 | 86.72 | 80.47 | 80.29 | 69.48 | 62.83 | 47.57 |
| Rade2021Helper_R18_ddpm | 86.86 | 81.12 | 75.67 | 74.39 | 64.92 | 57.09 | 43.99 |
| Rade2021Helper_R18_extra | 89.02 | 83.21 | 77.16 | 76.36 | 66.10 | 57.67 | 43.38 |
| Rebuffi2021Fixing_106_16_cutmix_ddpm | 88.50 | 84.32 | 78.88 | 78.83 | 70.59 | 64.64 | 52.31 |
| Rebuffi2021Fixing_28_10_cutmix_ddpm | 87.33 | 82.36 | 76.33 | 76.08 | 66.35 | 60.75 | 46.73 |
| Rebuffi2021Fixing_70_16_cutmix_ddpm | 88.54 | 84.31 | 78.64 | 78.79 | 68.95 | 64.25 | 49.75 |
| Rebuffi2021Fixing_70_16_cutmix_extra | 92.23 | 88.02 | 81.21 | 82.76 | 70.39 | 66.58 | 47.97 |
| Rebuffi2021Fixing_R18_ddpm | 83.53 | 77.99 | 71.46 | 71.68 | 61.83 | 56.66 | 44.26 |
| Rice2020Overfitting | 85.34 | 79.60 | 74.06 | 72.82 | 64.81 | 53.42 | 43.66 |
| Sehwag2020Hydra | 88.98 | 83.49 | 76.02 | 76.18 | 65.38 | 57.14 | 44.37 |
| Sehwag2021Proxy | 86.68 | 81.72 | 76.91 | 76.39 | 68.84 | 60.27 | 53.29 |
| Sehwag2021Proxy_R18 | 84.59 | 79.25 | 73.87 | 72.74 | 64.73 | 55.54 | 46.58 |
| Sehwag2021Proxy_ResNest152 | 87.21 | 82.55 | 78.08 | 77.30 | 70.78 | 62.79 | 56.67 |
| Sitawarin2020Improving | 86.84 | 80.21 | 74.16 | 72.47 | 63.54 | 50.72 | 43.25 |
| Sridhar2021Robust | 89.46 | 84.34 | 77.42 | 78.03 | 66.42 | 59.66 | 46.45 |
| Sridhar2021Robust_34_15 | 86.53 | 81.45 | 73.95 | 75.63 | 64.18 | 60.41 | 45.77 |
| Wang2020Improving | 87.51 | 82.15 | 74.59 | 75.47 | 63.17 | 56.29 | 42.57 |
| Wang2023Better_WRN-28-10 | 92.44 | 88.40 | 81.51 | 83.10 | 71.42 | 67.31 | 52.03 |
| Wang2023Better_WRN-70-16 | 93.26 | 89.72 | 82.88 | 84.90 | 73.95 | 70.69 | 55.17 |
| Wong2020Fast | 83.34 | 75.77 | 69.60 | 67.23 | 58.38 | 43.21 | 36.96 |
| Wu2020Adversarial | 85.36 | 79.69 | 73.33 | 72.90 | 62.64 | 56.17 | 42.28 |
| Wu2020Adversarial_extra | 88.25 | 82.98 | 76.83 | 76.61 | 66.45 | 60.04 | 46.70 |
| Zhang2019Theoretically | 84.92 | 78.96 | 71.68 | 71.15 | 60.30 | 53.08 | 40.05 |
| Zhang2019You | 87.20 | 79.42 | 72.40 | 70.29 | 60.61 | 44.83 | 36.65 |
| Zhang2020Attacks | 84.52 | 78.48 | 71.10 | 71.32 | 59.45 | 53.51 | 40.76 |
| Zhang2020Geometry | 89.36 | 84.01 | 81.57 | 77.10 | 73.45 | 59.64 | 53.50 |

Table 5: Complete list of $\mathcal{R}$ and its bounds under $(\mathcal{W}_\infty, l_2)$ threat model based on CE loss.

| | $\delta=1/8$ | | | $\delta=1/4$ | | |
|---|---|---|---|---|---|---|
| Network | $\mathcal{R}$ | $\Delta\mathcal{R}^u$ | $\Delta\mathcal{R}^l$ | $\mathcal{R}$ | $\Delta\mathcal{R}^u$ | $\Delta\mathcal{R}^l$ |
| Augustin2020Adversarial | 0.9649 | 0.0026 | -0.0153 | 0.9275 | 0.0083 | -0.0324 |
| Augustin2020Adversarial_34_10 | 0.9669 | 0.0035 | -0.0097 | 0.9303 | 0.0115 | -0.0201 |
| Augustin2020Adversarial_34_10_extra | 0.9752 | 0.0031 | -0.0150 | 0.9400 | 0.0084 | -0.0239 |
| Ding2020MMA | 0.9496 | 0.0016 | -0.0176 | 0.8912 | 0.0081 | -0.0434 |
| Engstrom2019Robustness | 0.9557 | 0.0019 | -0.0087 | 0.9069 | 0.0057 | -0.0256 |
| Gowal2020Uncovering | 0.9640 | 0.0022 | -0.0063 | 0.9271 | 0.0045 | -0.0191 |
| Gowal2020Uncovering_extra | 0.9779 | 0.0013 | -0.0121 | 0.9451 | 0.0050 | -0.0209 |
| Rade2021Helper_R18_ddpm | 0.9720 | 0.0017 | -0.0124 | 0.9339 | 0.0061 | -0.0209 |
| Rebuffi2021Fixing_28_10_cutmix_ddpm | 0.9724 | 0.0034 | -0.0107 | 0.9416 | 0.0060 | -0.0232 |
| Rebuffi2021Fixing_70_16_cutmix_ddpm | 0.9778 | 0.0021 | -0.0137 | 0.9489 | 0.0052 | -0.0259 |
| Rebuffi2021Fixing_70_16_cutmix_extra | 0.9795 | 0.0018 | -0.0091 | 0.9524 | 0.0040 | -0.0184 |
| Rebuffi2021Fixing_R18_cutmix_ddpm | 0.9686 | 0.0032 | -0.0150 | 0.9352 | 0.0061 | -0.0339 |
| Rice2020Overfitting | 0.9593 | 0.0030 | -0.0178 | 0.9061 | 0.0090 | -0.0342 |
| Rony2019Decoupling | 0.9517 | 0.0015 | -0.0144 | 0.8899 | 0.0077 | -0.0316 |
| Sehwag2021Proxy | 0.9724 | 0.0027 | -0.0099 | 0.9420 | 0.0065 | -0.0235 |
| Sehwag2021Proxy_R18 | 0.9703 | 0.0028 | -0.0133 | 0.9344 | 0.0057 | -0.0271 |
| Wang2023Better_WRN-28-10 | 0.9819 | 0.0013 | -0.0098 | 0.9541 | 0.0047 | -0.0147 |
| Wang2023Better_WRN-70-16 | 0.9814 | 0.0015 | -0.0069 | 0.9609 | 0.0029 | -0.0163 |
| Wu2020Adversarial | 0.9640 | 0.0024 | -0.0088 | 0.9270 | 0.0051 | -0.0221 |

Table 6: Complete list of $\mathcal{R}$ and its bounds under $(\mathcal{W}_2, l_2)$ threat model based on ReDLR loss.

| | $\delta=1/32$ | | | $\delta=1/16$ | | |
|---|---|---|---|---|---|---|
| Network | $\mathcal{R}$ | $\Delta\mathcal{R}^u$ | $\Delta\mathcal{R}^l$ | $\mathcal{R}$ | $\Delta\mathcal{R}^u$ | $\Delta\mathcal{R}^l$ |
| Augustin2020Adversarial | 0.9844 | 0.0058 | -0.0358 | 0.9723 | 0.0131 | -0.0505 |
| Augustin2020Adversarial_34_10 | 0.9868 | 0.0026 | -0.0518 | 0.9738 | 0.0094 | -0.0743 |
| Augustin2020Adversarial_34_10_extra | 0.9879 | 0.0049 | -0.0452 | 0.9756 | 0.0105 | -0.0652 |
| Ding2020MMA | 0.9788 | 0.0087 | -0.0694 | 0.9644 | 0.0148 | -0.1079 |
| Engstrom2019Robustness | 0.9834 | 0.0036 | -0.0798 | 0.9719 | 0.0085 | -0.1250 |
| Gowal2020Uncovering | 0.9837 | 0.0048 | -0.0581 | 0.9719 | 0.0110 | -0.0918 |
| Gowal2020Uncovering_extra | 0.9915 | 0.0026 | -0.0574 | 0.9855 | 0.0039 | -0.0898 |
| Rade2021Helper_R18_ddpm | 0.9879 | 0.0055 | -0.0530 | 0.9802 | 0.0075 | -0.0871 |
| Rebuffi2021Fixing_28_10_cutmix_ddpm | 0.9902 | 0.0034 | -0.0572 | 0.9826 | 0.0069 | -0.0912 |
| Rebuffi2021Fixing_70_16_cutmix_ddpm | 0.9920 | 0.0023 | -0.0610 | 0.9852 | 0.0055 | -0.0966 |
| Rebuffi2021Fixing_70_16_cutmix_extra | 0.9922 | 0.0020 | -0.0539 | 0.9851 | 0.0050 | -0.0852 |
| Rebuffi2021Fixing_R18_cutmix_ddpm | 0.9894 | 0.0034 | -0.0711 | 0.9791 | 0.0081 | -0.1104 |
| Rice2020Overfitting | 0.9859 | 0.0046 | -0.0825 | 0.9738 | 0.0104 | -0.1263 |
| Rony2019Decoupling | 0.9829 | 0.0045 | -0.0855 | 0.9672 | 0.0116 | -0.1267 |
| Sehwag2021Proxy | 0.9918 | 0.0018 | -0.0767 | 0.9824 | 0.0047 | -0.1165 |
| Sehwag2021Proxy_R18 | 0.9890 | 0.0039 | -0.0558 | 0.9789 | 0.0097 | -0.0868 |
| Wang2023Better_WRN-28-10 | 0.9902 | 0.0023 | -0.0492 | 0.9832 | 0.0054 | -0.0775 |
| Wang2023Better_WRN-70-16 | 0.9919 | 0.0020 | -0.0494 | 0.9850 | 0.0043 | -0.0780 |
| Wu2020Adversarial | 0.9867 | 0.0037 | -0.0704 | 0.9759 | 0.0092 | -0.1114 |

Table 7: Complete list of $\mathcal{R}$ and its bounds under $(\mathcal{W}_\infty, l_\infty)$ threat model based on CE loss.

| Network | $\delta=2/255$ | | | $\delta=4/255$ | | |
|---|---|---|---|---|---|---|
| | $\mathcal{R}$ | $\Delta\mathcal{R}^u$ | $\Delta\mathcal{R}^l$ | $\mathcal{R}$ | $\Delta\mathcal{R}^u$ | $\Delta\mathcal{R}^l$ |
| Addepalli2021Towards_RN18 | 0.9205 | 0.0172 | -0.0234 | 0.8314 | 0.0403 | -0.0390 |
| Addepalli2021Towards_WRN34 | 0.9361 | 0.0144 | -0.0266 | 0.8577 | 0.0374 | -0.0399 |
| Addepalli2022Efficient_RN18 | 0.9236 | 0.0126 | -0.0194 | 0.8327 | 0.0329 | -0.0312 |
| Addepalli2022Efficient_WRN_34_10 | 0.9352 | 0.0097 | -0.0167 | 0.8514 | 0.0275 | -0.0218 |
| Andriushchenko2020Understanding | 0.9054 | 0.0125 | -0.0200 | 0.8071 | 0.0296 | -0.0496 |
| Carmon2019Unlabeled | 0.9431 | 0.0056 | -0.0122 | 0.8682 | 0.0236 | -0.0144 |
| Chen2020Adversarial | 0.9261 | 0.0074 | -0.0319 | 0.8323 | 0.0206 | -0.0495 |
| Chen2020Efficient | 0.9199 | 0.0111 | -0.0107 | 0.8273 | 0.0293 | -0.0177 |
| Chen2021LTD_WRN34_10 | 0.9363 | 0.0099 | -0.0224 | 0.8564 | 0.0275 | -0.0347 |
| Chen2021LTD_WRN34_20 | 0.9382 | 0.0105 | -0.0280 | 0.8624 | 0.0322 | -0.0476 |
| Cui2020Learnable_34_10 | 0.9290 | 0.0070 | -0.0178 | 0.8410 | 0.0196 | -0.0324 |
| Cui2020Learnable_34_20 | 0.9267 | 0.0079 | -0.0150 | 0.8395 | 0.0224 | -0.0296 |
| Dai2021Parameterizing | 0.9468 | 0.0069 | -0.0173 | 0.8844 | 0.0139 | -0.0362 |
| Debenedetti2022Light_XCiT-L12 | 0.9440 | 0.0041 | -0.0154 | 0.8658 | 0.0162 | -0.0233 |
| Debenedetti2022Light_XCiT-M12 | 0.9435 | 0.0049 | -0.0164 | 0.8641 | 0.0194 | -0.0233 |
| Debenedetti2022Light_XCiT-S12 | 0.9428 | 0.0048 | -0.0239 | 0.8605 | 0.0198 | -0.0357 |
| Engstrom2019Robustness | 0.9231 | 0.0101 | -0.0203 | 0.8298 | 0.0263 | -0.0406 |
| Gowal2020Uncovering_28_10_extra | 0.9476 | 0.0072 | -0.0149 | 0.8803 | 0.0221 | -0.0214 |
| Gowal2020Uncovering_34_20 | 0.9323 | 0.0081 | -0.0156 | 0.8579 | 0.0206 | -0.0331 |
| Gowal2020Uncovering_70_16 | 0.9357 | 0.0055 | -0.0144 | 0.8614 | 0.0155 | -0.0279 |
| Gowal2020Uncovering_70_16_extra | 0.9535 | 0.0050 | -0.0129 | 0.8898 | 0.0196 | -0.0137 |
| Gowal2021Improving_28_10_ddpm_100m | 0.9528 | 0.0073 | -0.0192 | 0.8949 | 0.0177 | -0.0347 |
| Gowal2021Improving_70_16_ddpm_100m | 0.9551 | 0.0061 | -0.0147 | 0.9024 | 0.0169 | -0.0265 |
| Gowal2021Improving_R18_ddpm_100m | 0.9397 | 0.0065 | -0.0166 | 0.8726 | 0.0117 | -0.0387 |
| Hendrycks2019Using | 0.9340 | 0.0057 | -0.0250 | 0.8519 | 0.0215 | -0.0447 |
| Huang2020Self | 0.9294 | 0.0075 | -0.0139 | 0.8451 | 0.0270 | -0.0201 |
| Huang2021Exploring | 0.9471 | 0.0051 | -0.0091 | 0.8779 | 0.0227 | -0.0091 |
| Huang2021Exploring_ema | 0.9519 | 0.0054 | -0.0103 | 0.8856 | 0.0226 | -0.0103 |
| Huang2022Revisiting_WRN-A4 | 0.9537 | 0.0052 | -0.0101 | 0.8919 | 0.0167 | -0.0127 |
| Jia2022LAS-AT_34_10 | 0.9327 | 0.0102 | -0.0173 | 0.8524 | 0.0262 | -0.0277 |
| Jia2022LAS-AT_70_16 | 0.9372 | 0.0070 | -0.0178 | 0.8585 | 0.0229 | -0.0262 |
| Pang2020Boosting | 0.9319 | 0.0181 | -0.0349 | 0.8526 | 0.0430 | -0.0611 |
| Pang2022Robustness_WRN28_10 | 0.9449 | 0.0095 | -0.0199 | 0.8708 | 0.0229 | -0.0325 |
| Pang2022Robustness_WRN70_16 | 0.9524 | 0.0056 | -0.0220 | 0.8828 | 0.0225 | -0.0331 |
| Rade2021Helper_ddpm | 0.9444 | 0.0068 | -0.0195 | 0.8715 | 0.0188 | -0.0317 |
| Rade2021Helper_extra | 0.9481 | 0.0057 | -0.0158 | 0.8778 | 0.0186 | -0.0247 |
| Rade2021Helper_R18_ddpm | 0.9339 | 0.0098 | -0.0203 | 0.8564 | 0.0220 | -0.0407 |
| Rade2021Helper_R18_extra | 0.9347 | 0.0095 | -0.0183 | 0.8578 | 0.0220 | -0.0370 |
| Rebuffi2021Fixing_106_16_cutmix_ddpm | 0.9528 | 0.0063 | -0.0196 | 0.8907 | 0.0226 | -0.0295 |
| Rebuffi2021Fixing_28_10_cutmix_ddpm | 0.9432 | 0.0088 | -0.0193 | 0.8713 | 0.0262 | -0.0286 |
| Rebuffi2021Fixing_70_16_cutmix_ddpm | 0.9522 | 0.0071 | -0.0204 | 0.8899 | 0.0208 | -0.0311 |
| Rebuffi2021Fixing_70_16_cutmix_extra | 0.9544 | 0.0051 | -0.0169 | 0.8973 | 0.0191 | -0.0282 |
| Rebuffi2021Fixing_R18_ddpm | 0.9337 | 0.0078 | -0.0140 | 0.8584 | 0.0242 | -0.0270 |
| Rice2020Overfitting | 0.9327 | 0.0091 | -0.0273 | 0.8532 | 0.0199 | -0.0548 |
| Sehwag2020Hydra | 0.9383 | 0.0055 | -0.0123 | 0.8561 | 0.0262 | -0.0126 |
| Sehwag2021Proxy | 0.9428 | 0.0073 | -0.0150 | 0.8813 | 0.0166 | -0.0348 |
| Sehwag2021Proxy_R18 | 0.9369 | 0.0091 | -0.0209 | 0.8598 | 0.0240 | -0.0364 |
| Sehwag2021Proxy_ResNest152 | 0.9466 | 0.0062 | -0.0138 | 0.8864 | 0.0151 | -0.0308 |
| Sitawarin2020Improving | 0.9237 | 0.0084 | -0.0190 | 0.8345 | 0.0243 | -0.0428 |
| Sridhar2021Robust | 0.9428 | 0.0063 | -0.0112 | 0.8722 | 0.0221 | -0.0171 |
| Sridhar2021Robust_34_15 | 0.9413 | 0.0067 | -0.0065 | 0.8740 | 0.0208 | -0.0089 |
| Wang2020Improving | 0.9387 | 0.0119 | -0.0216 | 0.8624 | 0.0362 | -0.0303 |
| Wang2023Better_WRN-28-10 | 0.9563 | 0.0067 | -0.0149 | 0.8990 | 0.0166 | -0.0248 |
| Wang2023Better_WRN-70-16 | 0.9620 | 0.0049 | -0.0149 | 0.9104 | 0.0165 | -0.0239 |
| Wong2020Fast | 0.9093 | 0.0128 | -0.0282 | 0.8068 | 0.0301 | -0.0608 |
| Wu2020Adversarial | 0.9336 | 0.0068 | -0.0161 | 0.8540 | 0.0225 | -0.0267 |
| Wu2020Adversarial_extra | 0.9403 | 0.0073 | -0.0139 | 0.8681 | 0.0228 | -0.0230 |
| Zhang2019Theoretically | 0.9298 | 0.0061 | -0.0190 | 0.8381 | 0.0238 | -0.0289 |
| Zhang2019You | 0.9107 | 0.0073 | -0.0196 | 0.8061 | 0.0203 | -0.0506 |
| Zhang2020Attacks | 0.9285 | 0.0082 | -0.0167 | 0.8438 | 0.0257 | -0.0267 |
| Zhang2020Geometry | 0.9402 | 0.0187 | -0.0307 | 0.8629 | 0.0404 | -0.0514 |

Table 8: Complete list of $\mathcal{R}$ and its bounds under $(\mathcal{W}_2, l_\infty)$ threat model based on ReDLR loss.

| | $\delta=1/510$ | | | $\delta=1/255$ | | |
| Network | $\mathcal{R}$ | $\Delta\mathcal{R}^u$ | $\Delta\mathcal{R}^l$ | $\mathcal{R}$ | $\Delta\mathcal{R}^u$ | $\Delta\mathcal{R}^l$ |
|---|---|---|---|---|---|---|
| Addepalli2021Towards_RN18 | 0.9762 | 0.0080 | -0.1301 | 0.9533 | 0.0202 | -0.1902 |
| Addepalli2021Towards_WRN34 | 0.9774 | 0.0102 | -0.0849 | 0.9590 | 0.0197 | -0.1257 |
| Addepalli2022Efficient_RN18 | 0.9740 | 0.0088 | -0.1012 | 0.9534 | 0.0195 | -0.1480 |
| Addepalli2022Efficient_WRN_34_10 | 0.9749 | 0.0124 | -0.0645 | 0.9563 | 0.0231 | -0.0942 |
| Andriushchenko2020Understanding | 0.9679 | 0.0140 | -0.0969 | 0.9438 | 0.0294 | -0.1357 |
| Carmon2019Unlabeled | 0.9775 | 0.0079 | -0.0803 | 0.9584 | 0.0178 | -0.1162 |
| Chen2020Adversarial | 0.9639 | 0.0182 | -0.0351 | 0.9371 | 0.0351 | -0.0485 |
| Chen2020Efficient | 0.9713 | 0.0147 | -0.0742 | 0.9495 | 0.0281 | -0.1073 |
| Chen2021LTD_WRN34_10 | 0.9758 | 0.0140 | -0.0622 | 0.9578 | 0.0236 | -0.0960 |
| Chen2021LTD_WRN34_20 | 0.9780 | 0.0078 | -0.1077 | 0.9635 | 0.0124 | -0.1615 |
| Cui2020Learnable_34_10 | 0.9694 | 0.0121 | -0.0663 | 0.9462 | 0.0228 | -0.0838 |
| Cui2020Learnable_34_20 | 0.9710 | 0.0132 | -0.0722 | 0.9492 | 0.0231 | -0.0982 |
| Dai2021Parameterizing | 0.9814 | 0.0082 | -0.0825 | 0.9654 | 0.0164 | -0.1206 |
| Debenedetti2022Light_XCiT-L12 | 0.9760 | 0.0076 | -0.0730 | 0.9579 | 0.0181 | -0.1063 |
| Debenedetti2022Light_XCiT-M12 | 0.9781 | 0.0127 | -0.0477 | 0.9574 | 0.0280 | -0.0682 |
| Debenedetti2022Light_XCiT-S12 | 0.9795 | 0.0078 | -0.0911 | 0.9620 | 0.0161 | -0.1370 |
| Engstrom2019Robustness | 0.9746 | 0.0086 | -0.1117 | 0.9550 | 0.0174 | -0.1600 |
| Gowal2020Uncovering_28_10_extra | 0.9791 | 0.0075 | -0.0813 | 0.9623 | 0.0153 | -0.1213 |
| Gowal2020Uncovering_34_20 | 0.9748 | 0.0110 | -0.0786 | 0.9546 | 0.0220 | -0.1166 |
| Gowal2020Uncovering_70_16 | 0.9722 | 0.0162 | -0.0583 | 0.9552 | 0.0232 | -0.0906 |
| Gowal2020Uncovering_70_16_extra | 0.9801 | 0.0070 | -0.0786 | 0.9639 | 0.0136 | -0.1145 |
| Gowal2021Improving_28_10_ddpm_100m | 0.9814 | 0.0070 | -0.0852 | 0.9690 | 0.0119 | -0.1281 |
| Gowal2021Improving_70_16_ddpm_100m | 0.9838 | 0.0072 | -0.0778 | 0.9726 | 0.0114 | -0.1150 |
| Gowal2021Improving_R18_ddpm_100m | 0.9797 | 0.0069 | -0.1049 | 0.9626 | 0.0160 | -0.1520 |
| Hendrycks2019Using | 0.9778 | 0.0090 | -0.0963 | 0.9582 | 0.0183 | -0.1398 |
| Huang2020Self | 0.9714 | 0.0117 | -0.0764 | 0.9478 | 0.0252 | -0.1082 |
| Huang2021Exploring | 0.9770 | 0.0109 | -0.0556 | 0.9593 | 0.0208 | -0.0803 |
| Huang2021Exploring_ema | 0.9792 | 0.0091 | -0.0625 | 0.9623 | 0.0189 | -0.0923 |
| Huang2022Revisiting_WRN-A4 | 0.9810 | 0.0088 | -0.0521 | 0.9646 | 0.0181 | -0.0766 |
| Jia2022LAS-AT_34_10 | 0.9762 | 0.0139 | -0.0743 | 0.9575 | 0.0232 | -0.1118 |
| Jia2022LAS-AT_70_16 | 0.9751 | 0.0138 | -0.0707 | 0.9583 | 0.0219 | -0.1091 |
| Pang2020Boosting | 0.9745 | 0.0093 | -0.0244 | 0.9528 | 0.0195 | -0.0330 |
| Pang2022Robustness_WRN28_10 | 0.9823 | 0.0055 | -0.1311 | 0.9707 | 0.0095 | -0.1988 |
| Pang2022Robustness_WRN70_16 | 0.9836 | 0.0078 | -0.0882 | 0.9711 | 0.0139 | -0.1378 |
| Rade2021Helper_ddpm | 0.9808 | 0.0068 | -0.0936 | 0.9672 | 0.0124 | -0.1430 |
| Rade2021Helper_extra | 0.9809 | 0.0066 | -0.0846 | 0.9674 | 0.0130 | -0.1298 |
| Rade2021Helper_R18_ddpm | 0.9778 | 0.0056 | -0.1266 | 0.9627 | 0.0098 | -0.1904 |
| Rade2021Helper_R18_extra | 0.9784 | 0.0127 | -0.0720 | 0.9587 | 0.0244 | -0.1073 |
| Rebuffi2021Fixing_106_16_cutmix_ddpm | 0.9829 | 0.0046 | -0.0787 | 0.9681 | 0.0130 | -0.1163 |
| Rebuffi2021Fixing_28_10_cutmix_ddpm | 0.9816 | 0.0050 | -0.1001 | 0.9651 | 0.0115 | -0.1436 |
| Rebuffi2021Fixing_70_16_cutmix_ddpm | 0.9824 | 0.0072 | -0.0694 | 0.9680 | 0.0141 | -0.1035 |
| Rebuffi2021Fixing_70_16_cutmix_extra | 0.9809 | 0.0056 | -0.0718 | 0.9661 | 0.0126 | -0.1058 |
| Rebuffi2021Fixing_R18_ddpm | 0.9758 | 0.0104 | -0.0855 | 0.9564 | 0.0194 | -0.1276 |
| Rice2020Overfitting | 0.9773 | 0.0082 | -0.1061 | 0.9593 | 0.0159 | -0.1547 |
| Sehwag2020Hydra | 0.9758 | 0.0096 | -0.0805 | 0.9577 | 0.0192 | -0.1201 |
| Sehwag2021Proxy | 0.9818 | 0.0073 | -0.0831 | 0.9657 | 0.0162 | -0.1202 |
| Sehwag2021Proxy_R18 | 0.9795 | 0.0071 | -0.0964 | 0.9609 | 0.0154 | -0.1370 |
| Sehwag2021Proxy_ResNest152 | 0.9805 | 0.0144 | -0.0361 | 0.9667 | 0.0243 | -0.0573 |
| Sitawarin2020Improving | 0.9747 | 0.0088 | -0.1054 | 0.9511 | 0.0204 | -0.1505 |
| Sridhar2021Robust | 0.9766 | 0.0087 | -0.0832 | 0.9594 | 0.0160 | -0.1240 |
| Sridhar2021Robust_34_15 | 0.9749 | 0.0105 | -0.0602 | 0.9537 | 0.0221 | -0.0858 |
| Wang2020Improving | 0.9738 | 0.0195 | -0.0134 | 0.9552 | 0.0341 | -0.0185 |
| Wang2023Better_WRN-28-10 | 0.9840 | 0.0054 | -0.0767 | 0.9692 | 0.0129 | -0.1120 |
| Wang2023Better_WRN-70-16 | 0.9835 | 0.0049 | -0.0747 | 0.9727 | 0.0074 | -0.1121 |
| Wong2020Fast | 0.9674 | 0.0163 | -0.0901 | 0.9448 | 0.0268 | -0.1281 |
| Wu2020Adversarial | 0.9776 | 0.0089 | -0.0837 | 0.9577 | 0.0221 | -0.1215 |
| Wu2020Adversarial_extra | 0.9778 | 0.0077 | -0.0963 | 0.9588 | 0.0158 | -0.1419 |
| Zhang2019Theoretically | 0.9774 | 0.0132 | -0.0670 | 0.9565 | 0.0277 | -0.1003 |
| Zhang2019You | 0.9720 | 0.0135 | -0.1073 | 0.9502 | 0.0247 | -0.1580 |
| Zhang2020Attacks | 0.9724 | 0.0208 | -0.0225 | 0.9504 | 0.0386 | -0.0312 |
| Zhang2020Geometry | 0.9875 | 0.0035 | -0.1267 | 0.9777 | 0.0068 | -0.1838 |

