# OpenReview forum: "Wasserstein distributional robustness of neural networks"
_NeurIPS.cc/2023/Conference — NeurIPS 2023 poster_

### Official Review · Reviewer_1Lzk · 2023-07-04

**Soundness:** 4 excellent
**Presentation:** 4 excellent
**Contribution:** 3 good
**Rating:** 7
**Confidence:** 4

**Summary:**

This paper presents a novel adversarial attack methodology premised on Wasserstein distributionally robust optimization. This approach serves as a generalized technique that encompasses many established adversarial attack methods such as FGSM and TRADES. To ensure computational tractability, the authors introduce a first-order approximation of the attack loss. This approximation is particularly suited for gradient-based optimization methodologies, offering a much faster computational speed compared to SOTA techniques such as AutoAttack. Theoretical evidence is provided for certified bounds on adversarial accuracy and out-of-sample performance. Experiments on the CIFAR-10 dataset are conducted to demonstrate the effectiveness of the proposed method.

**Strengths:**

+ The proposed ReDLR loss (10) is both simple and effective. It is built on clear intuition, that the attack should perturb images classified far from the decision boundary more aggressively while leaving misclassified images unchanged. The theoretical analysis and guarantees provided further validate the proposed approach.

+ The method is presented using robust proofs and in-depth analysis. I didn't notice any substantial unjustifiable assumptions or logical inaccuracies. The section on "Bounds on Out-of-Sample Performance" is particularly intriguing. Despite substantial parts of the proof being left in the appendix due to page limits, the analysis still provides invaluable insights for future research on empirical distributions of test sets and broader applications of Wasserstein balls.

+ The empirical results demonstrate notable improvements over robust baseline methods like DLR. It is worth noting that while the proposed ReDLR method may appear very similar to DLR at first glance, the empirical results show significant improvements, attesting to the effectiveness of the theoretical analysis.

+ The paper is well-structured and clearly written. Even the first part of the method section could serve as a comprehensive introduction to DLR-based methods, making it approachable and comprehensible even for novices in the field.

**Weaknesses:**

- One weakness of the paper is that the number of experiments provided is limited (only results on CIFAR-10 with a limited number of baselines compared). However, considering the major contribution of the paper is to provide a comprehensive theoretical analysis and the existing results already demonstrate the effectiveness of the method, I would say the weakness of experiments does not affect the overall quality of the paper.


**Questions:**

- I personally found the intuition stated in line 171-173 is quite important to understand the method and should be further stressed, both in the introduction and the earlier subsections of the method.

- S^c seems to first appeare in line 215, but no clear definition is provided.

- How strong is Assumption 4.1? Though intuitively for classification problems it should be held, it would be great if the authors provide some experimental analysis (with several commonly used NNs), or pinpoint some references.

**Limitations:**

See Weakness and Questions.

---

> ### Author Rebuttal · Authors · 2023-08-08
>
> Thank you for your comments and praise. This is much appreciated. Thank you in particular, for emphasising our contributions in introducing the ReDLR loss function and results in the "Bounds on Out-of-Sample Performance" section which, while important, we were not able to stress appropriately ourselves. Your comment that "the analysis still provides invaluable insights for future research" was hugely rewarding to read. We answer below your questions and look forward to any further discussion.
>
> Re **Weaknesses** - thank you for your understanding. We were focused on delivering a theoretically complete paper with proper empirical illustration of the performance of our methods but we did not have time, nor space to be honest, to present a comprehensive analysis on multiple datasets. We have now addressed this and included, see our global rebuttal above, results for CIFAR-100 and ImageNet. Both showcase the effectiveness of our methods.
>
> You mention also a limited number of baselines compared. This relates to comments made by other reviewers who asked us to benchmark our attack against other attacks. We argue above, in the global rebuttal, against a direct comparison with papers focusing on AA methods and for using the AutoAttack benchmark instead. Nevertheless, we can reverse engineer a comparison setup ex-post. We do this for the CW attack proposed in paper [D] (see report 5QWe) as this was also requested by Reviewer 5QWe. Specifically, Table V in paper [D] reports the average budget needed for a successful attack (for Inception v3 on a subset of ImageNet) under $l_2$ and $l_\infty$ metrics. These averages can be thus seen as an upper bound on the $\mathcal W_1$ distance. We re-did their computationsusing the first 1000 correctly classified images from the ImageNet subset used by RobustBench. We obtained an average budget of $0.00617$ which agrees with their reported average of $0.006$ in Table V in [D]. We then computed the average of the squares which provides an upper bound on the $\mathcal W_2$ distance *squared*, and taking the square root gave us a budget of $0.00711$ to perform a distributional $(\mathcal W_2,l_\infty)$ attack. We used our proposed W-PGD(50)-ReDLR method. This resulted in 934 images being successfully attacked out of the 1000. The CW attack took ca 4.5h to run (on a Tesla V100-SXM2-32GB-LS GPU) and the time needed scaled *linearly* with the number of images as the attack has to be performed image by image (but with a very large variance of time per image). In contrast, our attack took ca 50min to run (on a NVIDIA A100-SXM4-40GB GPU). In addition, we use mini-batches so the computational speed can be improved in function of the GPU's available memory.
>
> Re **First Question** - thank you. We agree this needs to be stressed earlier on and more clearly. We will make this change.
>
> Re **Second Question** - thank you. We will introduce the notation for the complement of an event earlier on.
>
> Re **Third Question** - thank you for raising this. A discussion, even if brief, of this assumption should have been included. We provide details here and will add a short comment to the paper. We believe this is a reasonable assumption. First, it is an order of magnitude weaker than the $L$-smoothness assumption often made in the related literature see, e.g., Assumption B in Sinha et al. '18 (line 415) which requires the gradients $\nabla_x J$, $\nabla_\theta J$ to be $L$-Lipschitz in both $x$ and in $\theta$, see also Assumption 1 in Sridhar et al. '21 (arXiv:2106.02078) and the discussion therein, as well as references [24]-[27] therein, or Assumptions 1 and 2 in paper [B] listed by Reviewer 5QWe. In fact, in this Assumption 2 the authors in [B] required the $l_2$ operator norm of the Hessian of the loss function to be $L$-Lipschitz. Second, as $x\in [0,1]^n$ which is a compact, we note that any continuously differentiable function will be automatically Lipschitz.
>
> We note also that understanding the regularity of the gradient $\nabla_x J$ not only helps with checking Assumption 4.1 holds but also with answering other questions. One such question, raised by Reviewer 5QWe, is stability of our attack. Our one-step attack is given via an explicit formula, which involves the gradient and the reciprocal of $\Upsilon$, which in turn is an expected norm of the gradient over the test dataset. Stability of the attack will thus largely reduce to control on the smoothness of the gradient.
>
> Of course, in practice, we not only want Assumption 4.1 to hold but we also want the constant $L$ to be reasonable and known. Or more broadly we want to have a concrete handle of the smoothness of the gradient. This is more challenging, especially if we deal with a large dimensional model trained using many additional tricks and procedures. Here to our help comes a body of literature exploring empirically the loss landscape for trained networks and the impact of various training features on this landscape. We mention Li et al. '17 (arXiv:1712.09913) or Chromanska et al. '15 (arXiv:1412.0233) among the seminal papers. Applying this techniques to assess specifically the validity of Assumption 4.1 on particular trained models remains outside the focus of our paper.

---

> > ### Comment · Reviewer_1Lzk · 2023-08-17
> > **Re: Rebuttal**
> >
> > Thanks for the rebuttal. I will keep my positive rating.

---

> > > ### Author Response · Authors · 2023-08-19
> > >
> > > Thank you for getting back to us and for your support.

---

### Official Review · Reviewer_Q9Xf · 2023-07-06

**Soundness:** 4 excellent
**Presentation:** 3 good
**Contribution:** 3 good
**Rating:** 8
**Confidence:** 3

**Summary:**

The paper focuses on the issue of distributional robustness in deep neural networks and introduces a novel approach to address it. The authors present a tractable distributional attack within a Wasserstein ball, which allows attackers to perturb inputs in a non-uniform manner. They propose a first-order approximation of the problem and introduce a new loss function called ReDLR. This framework encompasses the Fast Gradient Sign Method (FGSM) attack as a special case. The authors also provide tractable bounds for adversarial accuracy against distributional threat models, which offer insights even for pointwise attacks. To validate their approach, experiments are conducted on the CIFAR-10 dataset using state-of-the-art deep neural networks on RobustBench, demonstrating the effectiveness of the attack and the tightness of the bounds.

**Strengths:**


The strengths of the paper are:

-The introduction of a tractable attack for Wasserstein Distributionally Robust Optimization (WDRO) stands as a significant contribution, as does the adversarial accuracy boundary. Both offer key advancements to the current understanding and applications of deep learning models under adversarial conditions.

-The paper is underpinned by robust mathematical foundations, and its place in relation to the existing body of work in the field is clearly and precisely articulated, demonstrating a deep and accurate understanding of the state-of-the-art.


**Weaknesses:**

-The guarantees provided in the paper are asymptotic in nature and depend on the quality of the first-order approximation of the problem. This introduces some limitations and potential uncertainty in the practical applicability of the results.

**Questions:**

The paper demonstrates strong mathematical foundations and is very well written with a very accurate state of the art. The contributions, particularly the formation of a tractable framework for Wasserstein Distributionally Robust Optimization (WDRO) with asymptotic guarantees, are significant.

However, certain concerns arise:

-The proposed attack and adversarial accuracy boundaries heavily rely on the validity of the first-order approximation. Therefore, one query would be how the first-order estimation scales with the dimension of the input space. Specifically, does this approach retain its efficacy when applied to larger vision problems, such as those found in ImageNet (224x224x3)?

-The use of the term 'certified bound' seems somewhat misleading as it actually refers to an asymptotic result. The proposed bounds, as they currently stand, cannot provide certification for a given radius (even a small one) as depicted in appendix Figure E1. Furthermore, the lower bound appears to be loose in the experiments. The approach proposed in the appendix to tighten the bound sometimes leads to overestimation, rendering it unsuitable for certification without additional guarantees. Therefore, how could this issue be addressed for a more accurate and reliable certification?



**Limitations:**

The authors adequately addressed the limitations

---

> ### Author Rebuttal · Authors · 2023-08-08
>
> Thank you for your praise of our contribution. In particular, your comments that our results offer "key advancements to the current understanding and applications of deep learning models under adversarial conditions" as well as demonstrating "a deep and accurate understanding of the state-of-the-art" were truly rewarding to read. We answer below your questions and look forward to any further discussion.
>
> Re **Weaknesses** - we agree, this is true for any methods relying on a first-order approximation. We acknowledged this further in the global response above. We believe the empirical results testify to practical relevance of our methods and we also suggest how combining our W-DRO attack with different number of steps could offer a practical way to assess the regime of applicability.
>
> Re **First Question** - we present now the empirical evidence on ImageNet. For *pointwise* threat our bounds are still very tight and perform similarly to the lower dimensional dataset CIFAR10. The computational times are likewise hundreds of times quicker than AutoAttack (note that AutoAttack is slower when the images are harder to attack and it has to deploy secondary attacks which likely explains why its on average slower for CIFAR-10 than for CIFAR-100).
> However, for the *distributional* threat the lower bound performs markably worse. We attribute this to more pronounced concavity of $V(\delta)$. Here, we see that performing 5 steps W-PGD-ReDLR attack offers a significant improvement on the one-step lower bound.
>
> Re **Second Question** - Thank you for this interesting question. We agree that *asymptotically certified bounds* might be a more appropriate term.
> We point out that in other contexts one also obtains certified bounds which might sometimes fail. For example, L. Li, T. Xie, and B. Li '23 (see line 326) or J. Cohen, E. Rosenfeld, and Z. Kolter '19 (line 340)  both derive *probabilistic certified robustness bounds* obtained from random smoothing. Those bounds only hold with a certain confidence. However the smoothing distribution is tuned, there exists a small probability that the derived certified bounds violate the truth. Even for small $\delta$ our methods incur an $o(\delta)$ error which could, in principle, lead to violations even if we have not seen these in our results on the three datasets.
>
> While most $\delta=8/255$ attacks may be imperceptible to human eyes, for us it does not count as a small-$\delta$ attack. In Figure 1, we plot the performance of the first order approximation for the W-DRO value. With delta=8/255, we see a roughly 20\% relative error and this means that, in practice, our bounds need to be improved at this level. This can be done using a W-PGD(n)-CE attack to better approximate the W-DRO value function and derive a tighter bound. We discussed this approach in some detail already in the related point in our global rebuttal. The approach offers a tradeoff between computational intensity and quality of the lower bound and was illustrated in Figure E.2. In addition we also applied it now to CIFAR-10 at $\delta=8/255$ to amend Figure E.1 and show the improved lower bounds using 5 iterations of PGD ($n=5$). We also carried out the approximation for $n=10$ and the change is negligible giving us confidence that using $\mathcal R^l(5)$ is justified.
>
> We acknowledge the approach outlined above is a practical answer offering a rule of thumb solution. A more comprehensive theoretical investigation would be challenging. One avenue would be to try to derive a second-order sensitivity for W-DRO problem and use it to bound the error of the first-order approximation. This would likely involve much stronger assumptions, in particular controlling the second derivatives, similarly to, e.g., Sinha et al. '18 (line 415) who assume the gradient of the loss function is $L$-Lipschitz. Another possibility would be to use duality approach combined with strong assumptions to control the convexity/concavity of the W-DRO function. Both directions appear to be in interesting but challenging and remain outside of the scope of the present study.

---

> > ### Comment · Reviewer_Q9Xf · 2023-08-18
> >
> > Thank you for addressing my query. I stand by my score and recommend accepting the paper.

---

> > > ### Author Response · Authors · 2023-08-19
> > >
> > > We are pleased to hear our answers successfully addressed your queries. Thank you for getting back to us and for your support.

---

### Official Review · Reviewer_5QWe · 2023-07-07

**Soundness:** 2 fair
**Presentation:** 3 good
**Contribution:** 2 fair
**Rating:** 4
**Confidence:** 4

**Summary:**

The paper proposes a new method combining Wasserstein distance, DRO, and adversarial attack. The novelty is that the method computes the adversarial attack bounded by the Wasserstein distance. The paper provides a bound on the performance of out-of-domain samples. It applies the new adversarial attack on CIFAR10 dataset. The numerical results show that the attack is effective on models that are robust to pointwise attacks, and computing attacks is much faster than other methods.

**Strengths:**

1. The paper proposes an interesting direction by combining DRO and adversarial attacks. It is clear in the writing and presenting methods.
2. The paper gives some theoretical results on the proposed methods, like lower bound on $R_{\delta}$. The paper also shows numerical results comparing the lower bound from different methods.
3. The paper provides numerical results to show that the adversarial attack is effective at attacking and fast to compute.

**Weaknesses:**

1. The paper misses some references and does not compare the method with the existing DRO adversarial attack methods, like [A].
[B, C] also have a similar approach by combining DRO with adversarial attacks. In the area of traditional attacks, the paper can compare with more methods, like CW attack in [D].

[A] Sinha, A., Namkoong, H., Volpi, R. and Duchi, J., 2017. Certifying some distributional robustness with principled adversarial training. arXiv preprint arXiv:1710.10571.
[B] Volpi, R., Namkoong, H., Sener, O., Duchi, J.C., Murino, V. and Savarese, S., 2018. Generalizing to unseen domains via adversarial data augmentation. Advances in neural information processing systems, 31.
[C] Hua, X., Xu, H., Blanchet, J. and Nguyen, V.A., 2022, December. Human imperceptible attacks and applications to improve fairness. In 2022 Winter Simulation Conference (WSC) (pp. 2641-2652). IEEE.
[D] Carlini, N. and Wagner, D., 2017, May. Towards evaluating the robustness of neural networks. In 2017 ieee symposium on security and privacy (sp) (pp. 39-57). Ieee.

2. The paper only does experiments on CIFAR10 dataset. It would be helpful to see results on other datasets.

**Questions:**

1. For the computation time in Table 2, do the authors compare with PGD? PGD is a simple gradient-based method, so the computation time should be very small. Other methods, like FGSM, are also very fast. Also, for Table 2, do all the algorithms stop when they successfully attack all the images?
2. In line 165, the authors mention another distributional attack, why this method is not compared to in the paper?
3. In line 39, the paper mentions the approach "is better suited for gradient-based optimization methods than duality approach adopted in most of the works to date". Can you clarify on this point?

**Limitations:**

The authors discuss some of the limitations of the work, including only valid on small-scale attacks. More limitations could include the stability of the attack and how it compares to more state-of-the-art attack methods. The method also should be compared with more attack methods, like DRO-based attacks. From the current experiments, it is not very convincing that the method outperforms many other adversarial attack methods.

---

> ### Author Rebuttal · Authors · 2023-08-08
>
> Thank you for your comments and questions which raise important points. While you thought our writing and presentation were clear, your questions allowed us to understand where we could have explained the key insights and contributions better, for which we are particularly grateful. We stride in this rebuttal to answer all your questions as well as to provide these additional clarifications.
>
> Re **Weakness 1** - thank you for mentioning these reference. [A] was a major pioneering work on distributional robustness which we cite, see lines 36, 70, 89, 158 for details. arXiv hosts 5 versions of the work, all with the same title, the first 4 have three authors whilst the newest 2020 version has R. Volpi added as a co-author. We felt it was best to cite the 2018 ICLR version of this work but we are happy to adjust the citation if required. More importantly, a comparison with all of the suggested papers is not straightforward as our focus is different. We discuss this in the global rebuttal above and argue that using AutoAttack is a better benchmark. These comments apply to comparison with [D] which developed the CW attack (please note also that this work was cited in our paper at line 55). Nevertheless, a reverse-engineered comparison is presented in Rebuttal to Reviewer 1Lzk.
>
> Both [B] and [C] utilize duality of DRO to conduct distributional adversarial attack with emphasis on the performance of neural networks on unseen distribution and  the human imperceptible attacks respectively. We are grateful for pointing these new references to us and we will  cite them as related works. However, neither of them show their attacks can attain or approximate the adversarial distribution within the given budget. In fact, their proposed algorithms amount to the DRO problem with a penalization term. Let us consider this in detail as it also answers **Question 3**. In [C], the W-DRO training is re-written using duality in equation (3). This requires a minimization over the Lagrangian $\lambda$. However, in their attack method [C] fixes $\lambda$ and simply uses a gradient descent on (4), see line 7 in their Algorithm 1. While their choice of $\lambda=1$ may be optimal for *some* budget $\delta$ there is no way to perform an optimal attack for a *given* budget $\delta$. Similarly, [B] change the W-DRO problem to (4) and perform a SGD on the surrogate loss $\phi_\gamma$ in (5).
> We believe, the same weakness applies to all methods using duality to re-write the W-DRO objective, e.g., Sinha et al. '18 (see ref at line 415), use duality in Prop. 5 but in Algorithm 1 a fixed $\gamma$ is used. In contrast, our one-step attack is an explicit first-order approximation to the optimal attack for a *given* budget $\delta$ and only involves $\nabla_x J$ and its norms which are fast to compute using standard methods. Note that the first step of (11) does *not* require any projection. That is only needed for multiple step version of the attack but can be done efficiently, see Appendix C for details.
>
> We hope our discussion clarifies the relation of our contribution to the SOTA in the field, and how our sensitivity approach leading to a first order approximation of the adversarial distribution is well-suited to gradient-based methods. We look forward to further discussion with you and hope to turn this weakness into a strength in your eyes.
>
> Re **Weakness 2** - we have now included the empirical study of performance of our methods on two other datasets: CIFAR100 and ImageNet, see the global rebuttal above. The results are fully in line with those obtained on CIFAR10 and testify, we believe, to general applicability of our methods.
>
> Re **Question 1** - In Table 2 we compare the computational time of the true robustness metric $\mathcal R_{\delta}$
> estimated using AutoAttack with the the computational time of our bounds. AutoAttack will stop if an image has been successfully attacked.
> Both bounds $\mathcal R^{l}$ and $\mathcal R^{u}$ have the same order of computation time as the FGSM attack since they are both gradient-based. As discussed above, we used AutoAttack as the benchmark as we are interested in estimating $\mathcal R_{\delta}$ as well as we can to illustrate the performance of our bounds. We did not compare with other attack methods here on their own. We can roughly view a PGD(50) attack as 50 times more expensive than the FGSM attack or than our bounds.
>
> Re **Question 2** - Staib and Jegelka '17 (:=[E]) is one of the earliest works considering adversarial attack under DRO framework. We believe, they introduced the *distributional* threat model and we cite them accordingly. However, they concentrate on adversarial training in the context of *pointwise* threat models, with limited consideration for distributional attacks. In fact, ReDLR loss and sensitivity of DRO are two key ingredients proposed in our manuscript which lead to effective distributional attacks. Indeed, we believe our paper is the first to design and perform AA to a given distributional budget. Our Table 1 offers an empirical validation of the claim in [E] about the relative strengths of the attacks. In addition, we believe our proposed W-PGD attacks can be potentially used for distributional adversarial training. This is mentioned in Appendix D but, but is an important project left for future research. Such a new training could be then meaningfully compared with the network trained in [E].
>
> Re **Question 3** - we addressed this above.
>
> Re **Limitations** - see above, the global rebuttal and our response to the last question of Reviewer 1Lzk. We stress that our aim is *not* to outperform an existing AA method and that a direct comparison is not meaningful as, to the best of our knowledge, no other attack method for a distributional threat with a given budget has been proposed in the literature. However, even for *pointwise* threat models, our contribution offers very efficient bounds on robustness and AutoAttack represent a SOTA benchmark.

---

### Official Review · Reviewer_fHjF · 2023-07-07

**Soundness:** 3 good
**Presentation:** 3 good
**Contribution:** 3 good
**Rating:** 6
**Confidence:** 4

**Summary:**

This paper links more general attacks with question of out-of-sample performance and Knightian uncertainty. To evaluate the distributional robustness of neural networks, this paper proposes a first-order AA algorithm and its multistep version. The proposed attack algorithms include Fast Gradient Sign Method (FGSM) and Projected Gradient Descent (PGD) as special cases. Furthermore, the authors provide a new asymptotic estimate of the adversarial accuracy against distributional threat models. The bound is fast to compute and first-order accurate, offering new insights even for the pointwise AA. It also naturally yields out-of-sample performance guarantees. This paper also conducts numerical experiments on the CIFAR-10 dataset to illustrate the theoretical results.

**Strengths:**

1. This paper proposes a unified approach to adversarial attacks and training based on sensitivity analysis for Wasserstein DRO. This approach leveraging results from Bartl et al. (2021) is better suited for gradient-based optimization methods than duality approach adopted in most of the works to date. This paper further links the adversarial accuracy to the adversarial loss, and investigate the out-of-sample performance.
2. This paper derives a general adversarial attack method. As a special case, this recovers the classical FGSM attack lending it a further theoretical underpinning. However, the proposed method also allows to carry out attacks under a distributional threat model which has not been done before. This paper develops certified bounds on adversarial accuracy including the classical pointwise perturbations. The bounds are first-order accurate and much faster to compute than the AutoAttack benchmark.

**Weaknesses:**

1. The Wasserstein distance for the adversarial robustness was usually discussed in the theoretical analysis. This paper utilizes the first order approximation of the adversarial loss on the CIFAR-10 for the verification. This approach contributes on the efficiency of the Wasserstein distributional robustness for the practical use. Yet, it is still questionable on the robustness on the large-scale dataset. Due to the computational complexity of the data dimension for the Wasserstein distance, it would be interesting to see the performance and the efficiency of the proposed approach on the image data such as tiny-ImageNet with dimension 64 and ImageNet with dimension 224.
2. The Wasserstein distance is about the overall loss rather than the single norm such as L_2 and L_\infty. Thus, the Wasserstein robustness may contribute to other types of robustness such as common corruptions, natural adversarial examples and the perturbations beyond the adversarial robustness. It would be more sufficient to see the potential improvement of the proposed approach on more general perturbed images.

**Questions:**

The proposed results are valid for small attacks and it is still questionable for the proposed approach for the practical use. Additionally, it would be more general for the proposed method to further discuss on the generalization ability of the Wasserstein-robust models.

**Limitations:**

This paper uses Wasserstein distributionally robust optimization (DRO) and obtain novel contributions leveraging recent insights from DRO sensitivity analysis. Yet, it would be better to clarify with more sufficient evidences. The detailed comments can be seen in the weaknesses.

---

> ### Author Rebuttal · Authors · 2023-08-07
>
> Re **Weakness 1** - We trust your first point was addressed by our new experiments showcasing the performance on CIFAR100 and ImageNet. In fact, while we agree that the complexity of Wasserstein distance in higher dim can cause troubles, this does not happen for our methods at all due to our formulae for the first order sensitivity. This means that the approximation to the worst case Wasserstein adversarial attack is explicit and only requires us to compute the gradient $\nabla_x J$. In consequence, the size of the image only affects the first layer of the DNN and the number of classes affects the last layer of the DNN. The computational time is thus mostly determined by the (hidden) architecture of the DNN model we consider. We hope that the performance and feasibility of our methods on these much larger datasets might turn this weakness into a new strength in your eyes.
>
> Re **Weakness 2** - Thank you for highlighting the potential of Wasserstein robustness on general perturbed images. While we focused mainly on adversarial attacks, our intended scope is indeed larger, as highlighted by the general discussion on Knightian uncertainty and the specific section on out-of-sample performance. We believe that the capacity of our methods to consider general image perturbations by viewing them as elements of an $l_2$ or $l_\infty$ ball is a strength of the DRO approach. Notwithstanding that, there are certain limitations we should highlight:
> * a) These perturbations might be relatively large implying radius of perturbation ball outside of the linear approximation regime to which our methods are adapted.
> * b) The relevance of our bounds for some perturbation contexts might be limited. For instance, while our analysis targets worst-case adversarial scenarios, common corruptions focus on random real-world perturbations not specifically crafted to deceive neural networks. These corruptions are inherently non-adversarial, model-agnostic, and predefined. Hence, we expect model performance against such corruptions to surpass our derived bounds, limiting their insights under these scenarios.
>
> Both a) and b) could potentially be addressed by changing the underlying metric on images and using a bespoke one instead of $l_2$ or $l_\infty$.
> We note that paper [C] listed by Reviewer 5QWe used a bespoke metric designed to capture if two images were distinguishable to a human eye to better describe (classical) adversarial attacks. We believe this idea could be applied in other contexts as well.
> Specifically, to capture common corruptions, we would need to implement a metric which would measure how close one image is to being a common corruption of the other (say as defined by ImageNet-C and 3DCC). If we used this metric in defining the Wasserstein distance, common corruptions of our test dataset would all lie in a small ball around it. We could then, in principle, establish (asymptotically) certified bounds for distributional common corruption attacks, following the methodology presented in our paper. However, designing such metrics meeting the prerequisites for subsequent derivations (e.g., for our methods it needs to come from a semi-norm and we need to be able to compute its dual) is not trivial and falls outside the scope of our current work. We thank you for your suggestion and see it as a promising direction for future works.
>
> Re **Question** - the question of validity of the asymptotic regime was raised by another reviewer as well and is discussed in our general response above. As for generalisation ability, our methods can naturally provide insights (and bounds) on performance on unseen data generated from the same (or Wasserstein-close) distribution as the training data. We highlight this in the section *Bounds on out-of-sample performance* and in *Corollary 5.3* in particular. These tools apply to any models. If we understand well the last part of this question, the reviewer believes it would be interesting to investigate specifically how well Wasserstein-DRO-*trained* models perform on this criterion, and indeed compare their performance with other robust models. We empathically agree. While clearly outside the scope of the current paper, we hope to investigate this subject in a subsequent work. Indeed, in Appendix D, we suggested a natural Wasserstein-DRO-training method resulting from our insights and these will be the natural starting point for such further research.
>
> Re **Limitations** - we of course agree and have now included the empirical study of performance of our methods on two higher-dimensional datasets. The results are fully in line with those obtained on CIFAR10 and testify, we believe, to general applicability of our results.

---

> > ### Comment · Reviewer_fHjF · 2023-08-19
> >
> > Thanks for the detailed response. The authors have provided the convincing explainations for the weaknesses of the proposed approach such as the general perturbed images. I agree with the mentioned limitations to some degree. The authors also provided the empirical evidences in higher dimensions. Thus, I would raise my score.

---

> > > ### Author Response · Authors · 2023-08-19
> > >
> > > Thank you for getting back to us. We tried our best to address all the questions and queries comprehensively. We are therefore very pleased to hear you found our rebuttals detailed and convincing. If you have any remaining questions or doubts please let us know and we would be very glad to engage in a discussion.

---

### Author Rebuttal · Authors · 2023-08-07

We are deeply grateful to all the reviewers for their comments and questions. We were pleased to hear the reviewers felt our paper was well structured and clearly presented. It was rewarding to see all the reviewers list numerous strengths of our work and very interesting that their emphasis was different, testifying to the multi-faceted nature of our contribution. You felt we offered ''key advancements to the current understanding and applications of deep learning models under adversarial conditions'' and provided ''invaluable insights for future research''. This is exciting to hear. Equally, your questions were often eye-opening for us, thank you. We address some of them here and the rest in the individual replies. We look forward to discussing these with you.

**Performance of our methods on other datasets and in higher dimensions**.
The reviewers requested to see the performance of our methods on a higher-dimensional dataset. In the attached PDF, we illustrate the performance on CIFAR-100 and ImageNet datasets. We analyse all the models included on RobustBench leaderboards. For ImageNet, we use the 5000 images selected by RobustBench. The plots and the table mirror those for CIFAR-10 in our submission. For pointwise threat models, our upper and lower bounds sandwich the true (AutoAttack) accuracy ratio very well at a fraction of the computational cost. For distributional threat models, our bounds work but the tightness of the bound is model-dependent. We display also the improved bound obtained with 5 steps of W-PGD attack which leads to a marked improvement in the lower bound's performance. This is akin to Figure E.2 in our supplementary material (aka Appendix). We only consider the $l_\infty$ norm on images as RobustBench does not provide any models for $l_2$ for these higher-dim datasets.

**Asymptotic character of our results and their validity regime**.
To draw a parallel with calculus: linear approximations to a function offer countless applications, indeed are the basis of any gradient descent method, but are typically only useful locally and have to be handled with care. Similarly, we believe that our methods bring new powerful insights to the W-DRO robustness of NN. This was recognised in many of the reviewers' comments. Equally, the methods have their limitations which we acknowledge. We believe that the empirical results we presented from all the leaderboard models on RobustBench across CIFAR-10, CIFAR-100 and ImageNet and across different attack types provide an overwhelming evidence that our results are valid and relevant for the range of attack budget $\delta$ considered in AA settings. However, we do not have theoretical results to provide guarantees on the range of applicability.

In practice, we would compare our one-step-attack bound $\mathcal R^l$ with the bound obtained by iterating the attack several steps: we propose W-PGD(5) in our paper and report $\mathcal R^l(5)$. If the difference between the two is small it indicates linear approximation is working very well. If the difference is significant, we would use $\mathcal R^l(5)$ and maybe compare it against $\mathcal R^l(10)$. If we observe that $V(\delta)$ is convex - as for the CE loss under $(\mathcal W_\infty, l_\infty)$ attack - the lower bound should decrease. In this case, the one-step bound may in fact not be a lower bound for $\delta$ too large as shown in Figure E.1 in the Appendix. If we observe $V(\delta)$ is concave - as for the ReDLR loss and our $\mathcal W_2$ attacks - the one-step lower bound might be too low and will increase with additional PGD steps. This is visible in Figure E.2 and in the attached PDF for CIFAR-100 and ImageNet results, and for CIFAR-10 with $\delta=8/255$.

**Comparison with existing adversarial attack (AA) methods and the advantages of our approach**. The focus of our contribution is different to many previous works proposing an adversarial attack (AA) method. The latter often focuses on reporting the minimal budget needed for a successful attack - the methods are designed to continue attacking until successful and report the distortion level they required. In contrast, we are interested in fixing the attack budget upfront and computing (and doing so fast) how robust a given network is to attacks within that budget. This makes a comparison difficult as the purpose is different. Nevertheless, a reversed-engineered comparison is included in Rebuttal to Reviewer 1Lzk below.

This difficulty is solved by taking AutoAttack as a reference *pointwise* attack method as it works to a specified attack budget.
Hence, we argue that for *pointwise* threat models it suffices to compare with AutoAttack and we use it for canonical deltas on RobustBench. AutoAttack is an ensemble of white-box and black-box attacks including Auto-PGD based on CE loss and DLR loss. In Corce 20 [Section 4.2], the authors claim that DLR loss is more stable than CW loss with less severe failure of attacks. We thus feel that using AutoAttack, if indirectly, includes a comparison to the CW attack which hopefully answers Reviewer 5QWe's point.
We emphasize that the aim of our paper is to open the window of Wasserstein distributional attack and to provide computationally efficient generic way to compute bounds on robustness. Our focus is not to outperform specific existing adversarial attacks for *pointwise* threat models.

W-DRO formulation has been explored in a number of works, as discussed. However, we believe, none of them proposed an AA method for the associated distributional threat and a *given* attack budget. Hence there is no direct competition for us to compare against. This highlights the strength of our approach using sensitivity analysis as contrasted with dual formulation exploited in the earlier works. We discuss this in more detail in our rebuttal to Reviewer 5QWe below. This point was also stressed by Reviewer 1LzK who also asked us to emphasise this more clearly, which we will do.

---

### Decision · Program_Chairs · 2023-09-21

**Decision:**

Accept (poster)

**Comment:**

The paper presents a new approach by integrating Distributionally Robust Optimization (DRO) with adversarial attacks, significantly enhancing the field's understanding of neural network robustness. By introducing a first-order Adversarial Attack (AA) algorithm, the authors not only encompass existing methods like FGSM and PGD but also offer new insights into distributional threat models. The paper's strength lies in its solid theoretical underpinning and the introduction of first-order accurate bounds on adversarial accuracy, validated by numerical experiments on the CIFAR-10 dataset. This work efficiently bridges the gap between adversarial attacks and out-of-sample performance, and is especially noteworthy for its fast computation relative to existing benchmarks. Given these merits, I recommend accept.